# Quantifying spatio-temporal variation in aquaculture production areas in Satkhira, Bangladesh using geospatial and social survey

Hafeza Nujaira[1], Kumar Arun Prasad[2], Pankaj Kumar[3], Ali P. Yunus[4,5], Ali Kharrazi[6,7], L. N. Gupta[8], Tonni Agustiono Kurniawan[9], Haroon Sajjad[10], Ram Avtar[1,11]*

1 Graduate School of Environmental Science, Faculty of Environmental Earth Science, Hokkaido University, Sapporo, Japan, 2 Department of Geography, School of Earth Sciences, Central University of Tamil Nadu, Thiruvarur, Tamil Nadu, India, 3 Adaptation and Water, Institute for Global Environmental Strategies, Hayama, Japan, 4 Center for Climate Change Adaptation, National Institute for Environmental Studies, Tsukuba, Japan, 5 Department of Earth and Environmental Sciences, Indian Institute of Science Education and Research Mohali, Punjab, India, 6 Advanced Systems Analysis Group, International Institute for Applied Systems Analysis, Schlossplatz, Laxenburg, Austria, 7 Global Studies Program, Akita International University, Yuwa City, Akita, Japan, 8 Central Pollution Control Board, East Arjun Nagar, Delhi, India, 9 Key Laboratory of the Coastal and Wetland Ecosystems, Xiamen University, Xiamen, China, 10 Department of Geography, Faculty of Natural Sciences, Jamia Millia Islamia, New Delhi, India, 11 Faculty of Environmental Earth Science, Hokkaido University, Sapporo, Japan

* ram@ees.hokudai.ac.jp

**Data Availability Statement:** All relevant data are available on Zenodo: https://zenodo.org/record/6946963#.Y3BRcHZBy8g.

## Abstract

Despite Bangladesh being one of the leading countries in aquaculture food production worldwide, there is a considerable lack of updated scientific information about aquaculture activities in remote sites, making it difficult to manage sustainably. This study explored the use of geospatial and field data to monitor spatio-temporal changes in aquaculture production sites in the Satkhira district from 2017–2019. We used Shuttle Radar Topographic Mission digital elevation model (SRTM DEM) to locate aquaculture ponds based on the terrain elevation and slope. Radar backscatter information from the Sentinel-1 satellite, and different water indices derived from Sentinel-2 were used to assess the spatio-temporal extents of aquaculture areas. An image segmentation algorithm was applied to detect aquaculture ponds based on backscattering intensity, size and shape characteristics. Our results show that the highest number of aquaculture ponds were observed in January, with a size of more than 30,000 ha. Object-based image classification of Sentinel-1 data showed an overall accuracy above 80%. The key factors responsible for the variation in aquaculture were investigated using field surveys. We noticed that despite a significant number of aquaculture ponds in the study area, shrimp production and export are decreasing because of a lack of infrastructure, poor governance, and lack of awareness in the local communities. The result of this study can provide in-depth information about aquaculture areas, which is vital for policymakers and environmental administrators for successful aquaculture management in Satkhira, Bangladesh and other countries with similar issues.

**Funding:** The first author would like to thank JEES, Docomo Scholarship Foundation, for providing scholarship. This work was partially funded by Asia Pacific Network for Global Change Research (APN) under Collaborative Regional Research Programme (CRRP) with project reference number CRRP2019-01MY-Kumar.

**Competing interests:** The authors have declared that no competing interests exist.

# 1. Introduction

Aquaculture is vital for global food supply and one of the fastest-growing food production sectors. It provides about 15% of animal protein intake for 4.3 billion people worldwide, 80% of which is produced in Asia [1, 2]. Although inland and marine fish capture meets half of the global fish demand, the contribution of aquaculture to the global fish supply has increased steadily from 25.7% in 2000 to 46.8% in 2016 [3, 4]. Although aquaculture has many positive impacts, such as supporting local people's livelihoods, eliminating poverty, promoting the rural economy and improving food security; widespread aquaculture still has some negative consequences as well, for example, destruction of the ecosystem, increasing soil and water salinity for the longer term, waterlogging, change in water quality, decrease in rice or other crop production, etc. [5].

Bangladesh possesses a large deltaic environment, part of the Ganga-Brahmaputra-Meghna basin, and hence are highly favourable for fisheries and aquaculture production [6]. Additionally, Bangladesh is ranked 3rd in inland fish production worldwide after China and India, 5th in aquaculture production, and 11th in marine fish production in 2018 [7]. The fish and its related products are exported to around 60 countries around the world, especially to the European Union (EU), the USA and Japan [7]. This sector provides vast employment, food security, and socio-economic growth. Approximately 18 million people are directly and indirectly involved in the fish and aquaculture sector, while 1.4 million women are involved in this sector for their livelihoods by participating in the activities of fishing, cultivation, harvesting and processing [7]. The fishery sector contributes around 4.43% to the GDP of Bangladesh. In the last two decades, fish production has grown significantly, that is, from 1,781 million metric tons in 2000–2001 to 4,134 million metric tons in 2016–2017 [8]. In the last ten years, an average annual growth rate of nearly 5.43% has been observed [9]. The value of fish exports has increased roughly from 168 million USD in 1990 to 592.5 million USD in 2012 [10].

Despite the high economic revenue and benefits of aquaculture, there are many challenges related to getting reasonable pricing in international markets due to poor governance and management system in developing nations like Bangladesh. This scenario urges both scientific communities and policy makers to put more effort into research activities for sustainable aquaculture [11, 12]. Diligent monitoring is essential to detect changes in the extent of aquaculture areas over time using different satellite data in time series and various associated index values, which will be helpful in identifying sustainable management strategies [13–17].

Around the world, different methodologies have been used to monitor spatio-temporal variation in aquaculture areas and its environmental effects [18], changes in land use land cover pattern around aquaculture/mangroves [19], changes in biodiversity around aquaculture [20], changes in water quality in and around aquacultural areas [21], application of geospatial tools for monitoring fisheries sector [22] etc. Out of these methodologies, the first and foremost important is to monitor the trend of aquaculture area using various remote sensing imageries (like multi-spectral, Synthetic Aperture Radar satellite (SAR) etc.) and this can provide a vital tool in this direction. Ottinger et al. [23] mapped aquaculture ponds in Vietnam and China using data from SPOT-5, WorldView-1 and Sentinel-1 and applied object-based image analysis methods. Xia et al. [15] observed aquaculture ponds in China using Sentinel-1 and Sentinel-2 data. They identified aquaculture areas using the Normalized Difference Water Index (NDWI), Modified Normalized Difference Water Index (MNDWI), and Automated Water Extraction Index (AWEI) with different threshold values and also applied random forest classifiers. Duan et al. [16] mapped aquaculture ponds and salt fields using Landsat 5, and Landsat 8 imageries by applying the AWEInsh, MNDWI, and Land-use dynamic index (LUDI). Prasad et al. [17] observed aquaculture ponds in India using Sentinel-1 and very high-resolution

(VHR) Pleiades imagery by applying connected component segmentation with object-based image filtering.

Although Bangladesh plays an important role in the aquaculture sector and contributes significantly to the national GDP, the lack of up-to-date, explicit and continuous spatial knowledge about aquaculture imposes a great hurdle in its sustainable management. Here, we employed multi-temporal Sentinel-1 and Sentinel-2 images from 2017–2019 to track the changes in the aquaculture area in the Satkhira district, Bangladesh. This study focuses on providing a holistic picture of factors responsible for the trend in aquaculture, its related opportunities and challenges for people in the Satkhira district. The main objectives of this study are: (a) Monitor the spatio-temporal extent of aquaculture ponds from 2017 to 2019 in Satkhira district using an integrated geospatial and field approach; and (b) provide detailed information on socio-economic perspectives on aquaculture in Satkhira, Bangladesh, to enable more sustainable and profitable management. While the first objective can be achieved by a quantitative remote sensing approach, the second objective is qualitative and used key informant interviews with the relevant stakeholders in the region.

This study will be useful to identify not only the spatio-temporal variation but as well problems associated with the aquaculture industry. It also proposes a possible management solution that can be beneficial for common farmers and other stakeholders, such as the government and NGOs. In addition, this research will play an important role for the government in achieving the SDG goals. In particular, mapping and quantification of existing aquaculture areas can contribute to food security (SDG 2); clean water and sanitation (SDG 6.0), economic growth and better livelihoods (SDG 8); and sustainable consumption (SDG 12), to name a few.

## 2. Study area

Satkhira district is located in the south-western part of Bangladesh and a part of Khulna Division with coordinates 22.68˚ N latitudes and 89.07˚E longitudes [24]. The total area of this district is 3,858.33 km$^2$. This study focuses on five out of seven subdistricts under the Satkhira district, *viz*. Satkhira Sadar, Assasuni, Debhata, Tala and Kaligange (Fig 1). Most of the people in the study area depend on pisciculture or aquaculture, locally called *gher*. For aquaculture, freshwater is available in the Satkhira Sadar and Tala subdistricts, while brackish water is available in the Kaligange, Assassuni and Debhata subdistricts (Table 1). Although, the total cultivable land area is 229,607 ha, more than half i.e. 153,110 ha land area is covered with saline land. The total number of fish farmers in the Satkhira district is reported to be 76,394 who are directly involved in aquaculture. On average, 133,325 metric tons (MT) of shrimp and different types of fish are produced annually, among them, 87,777 MT of product is exported within and outside of the country [25]. Among the different varieties of shrimp that are cultivated in Satkhira, the most famous is the black tiger *Penaeus monodon* (locally called Bagda), which is grown in brackish water. The giant freshwater prawn, *Macrobrachium Rosenbergii*, (locally called Galda) is grown in freshwater. Table 1 gives an overview of shrimp farming in the study area.

## 3. Materials and methods

We used Sentinel-1 SAR images, Sentinel-2, optical images and SRTM DEM to extract information about the aquaculture area. It is possible to distinguish aquaculture areas from other water bodies such as rivers, lakes, and reservoirs based on backscattering and shape/size information [23]. Additionally, we collected secondary data from different government offices and conducted a questionnaire survey during the field visit in December 2019. Fig 2 shows the flowchart of the methodology used to extract aquaculture areas. The Google Earth Engine©

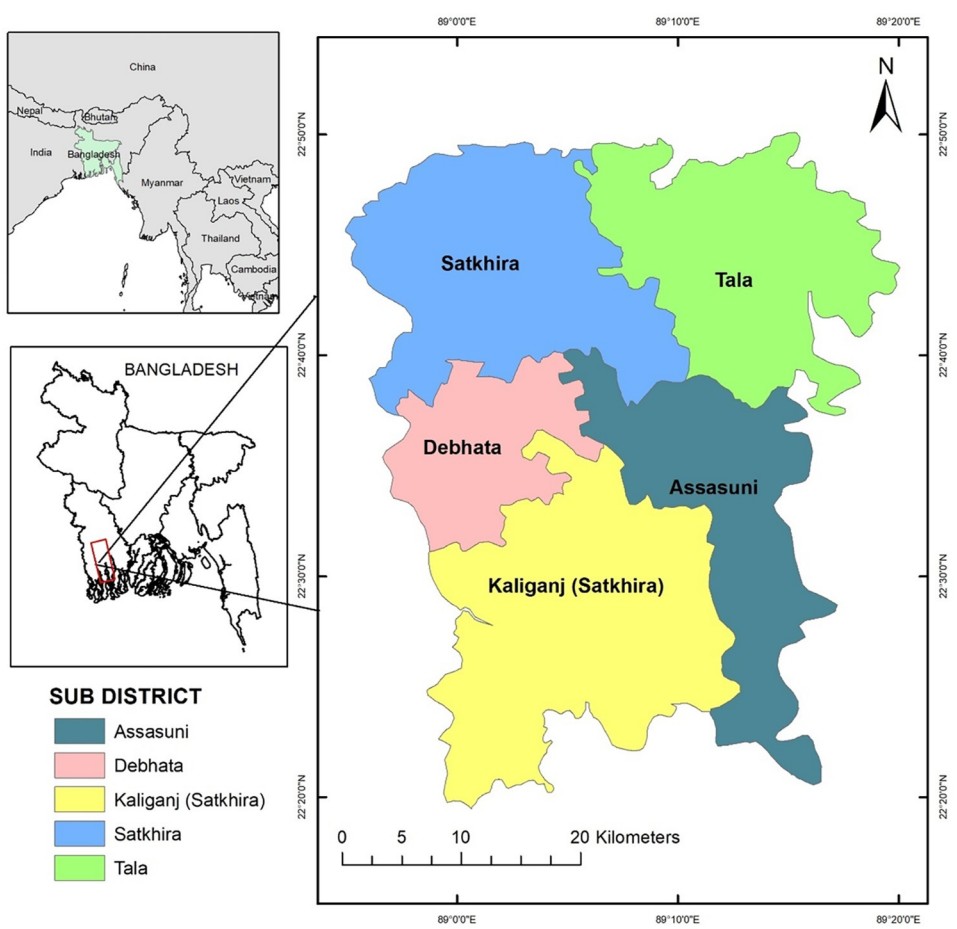

**Fig 1. Location map of the study area, Satkhira, Bangladesh.**

platform was used to download all the satellite data. Further post-processing and statistical analysis was performed with the help of ENVI, ArcGIS, and Orfeo toolbox to extract various thematic information.

### 3.1. Remote sensing data

**3.1.1. Sentinel-1 (SAR) data.**   In this study, dual-polarized multitemporal Sentinel-1 (VV + VH) data from January 2017 to December 2019 (descending mode) were used in interfero-metric wide width (IW) mode and ground-range detected high resolution (GRDH) format.

**Table 1. Overview of shrimp farming in the study area.**

| No. | Sub-district | Area (km$^2$) | Water type | Common shrimp species |
|---|---|---|---|---|
| 1 | Satkhira Sadar | 371.20 | Fresh water | *Macrobrachium Rosenbergii* (Galda) |
| 2 | Tala | 332.33 | Fresh water | *Macrobrachium Rosenbergii* (Galda) |
| 3 | Assassuni | 273.95 | Brackish water | *Penaeus monodon*, (Bagda) |
| 4 | Kaligange | 441.64 | Brackish water | *Penaeus monodon*, (Bagda) |
| 5 | Debhata | 171.97 | Brackish water | *Penaeus monodon*, (Bagda) |

Data source: Secondary data from district fisheries office (2017–2019)

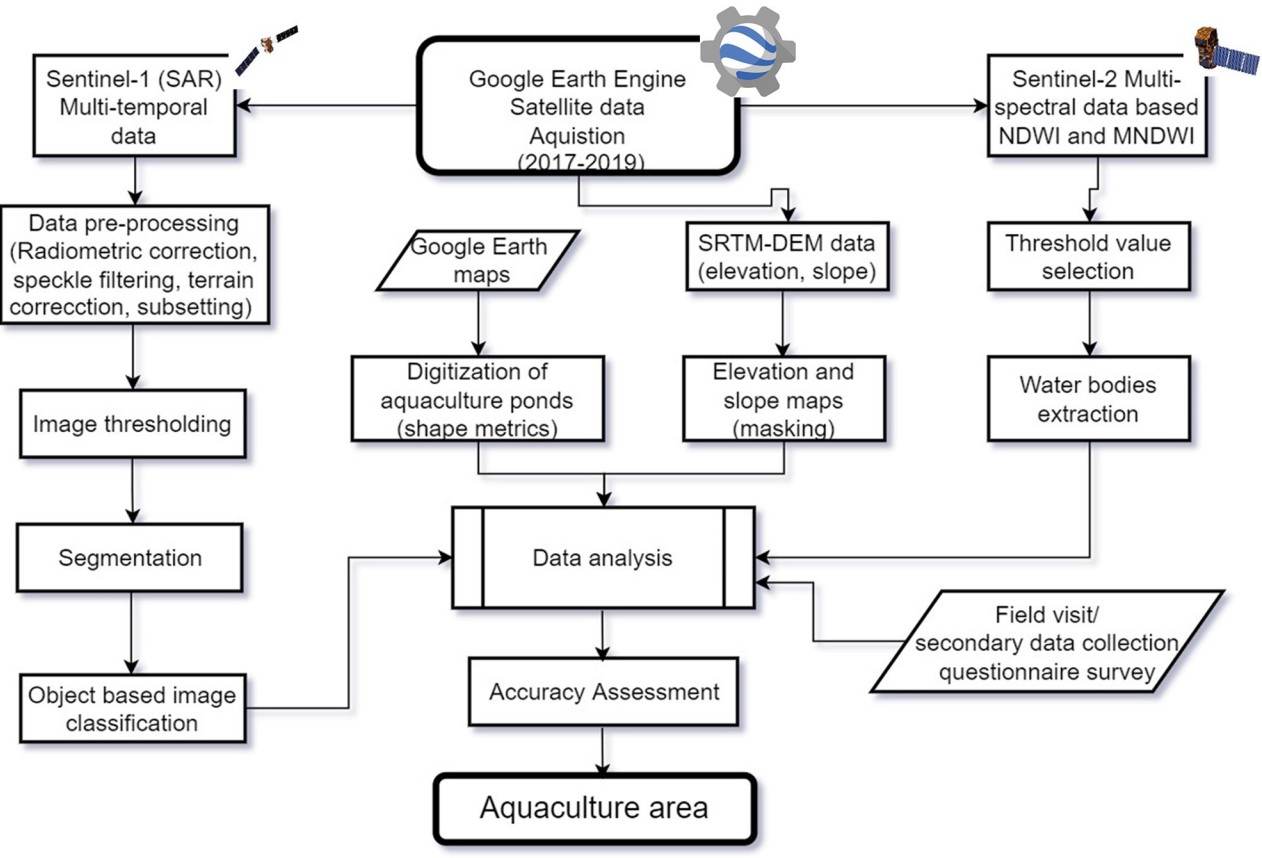

**Fig 2. Workflow for aquaculture area extraction in this study.**

Sentinel-1 is a series of two satellites (Sentinel 1A and Sentinel 1B), having a SAR instrument aboard operating at C-band of 5.5 GHz frequency and delivering at a resampled spatial resolution of 10 m. Google Earth Engine (GEE) platform was used for Sentinel-1 data acquisition. The Sentinel-1 GEE collection includes Ground Range Detected (GRD) scenes, processed using the Sentinel-1 Toolbox to generate a calibrated, ortho-corrected product. A total of 56 Sentinel-1 scenes available between 2017 January and 2019 December were used in this study (See S1 Table). Each scene was pre-processed with Sentinel-1 Toolbox using the following steps: (i)Thermal noise removal, (ii) Radiometric calibration and (iii) Terrain correction using SRTM 30. The final terrain-corrected values are converted to decibels via log scaling (10*log10 (x)). This pre-processed Sentinel-1 data was further used to classify based on object-based image classification in the Orfeo toolbox.

**3.1.2. Sentinel-2 data.** The Copernicus Sentinel-2 mission comprised a constellation of two satellites (Sentinel 2A and Sentinel 2B) and was started in June 2015 by the European Space Agency (ESA). This mission aimed to monitor the variability on the earth's surface. We obtained the pre-processed surface reflectance Sentinel-2 L2A data from GEE through scihub. They were initially computed by running Sen2Cor processor, consists of scene classification and atmospheric correction applied to Level-1C orthoimage product. Atmospheric correction in Sen2Cor is performed using a set of look-up tables generated via libRadtran. Baseline processing is the rural/continental aerosol type. The aerosol type and visibility or optical thickness of the atmosphere is derived using the Dense Dark Vegetation (DDV) algorithm. Clouds if any present in the scene are removed by using COPERNICUS/S2_CLOUD_PROBABILITY

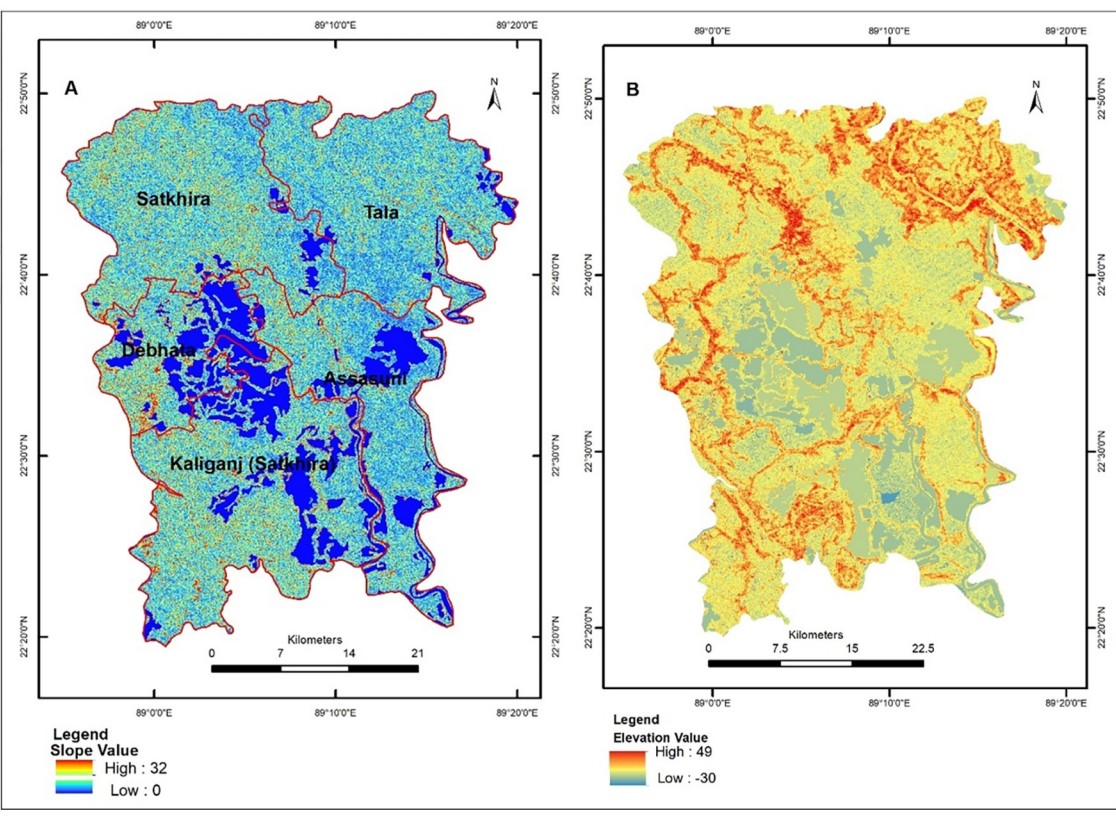

**Fig 3.** SRTM DEM data based (A) slope (B) elevation of the study area.

algorithm. A total of 380 Sentinel-2 scenes available between 2017 January and 2019 December were processed in GEE for NDWI and MNDWI extraction (See S1 File). The surface reflectance product of Sentinel-2 satellite is useful in land cover mapping with 13 spectral bands and spatial resolution varies from 10m to 60m depending on the spectral band. For monitoring and detecting open water bodies, three bands, B3 (Green), B8 (Near Infrared), B11 (Short wave infrared) of Sentinel-2 were used to derive indices such as NDWI and MNDWI as supported by previous studies [26–29].

**3.1.3. SRTM DEM data.** The open-source SRTM DEM data with 30m resolution were used for terrain masking to extract terrain details (elevation and slope) and assess possible aquaculture areas within the study area. The void-filled data was downloaded from the United States Geological Survey (USGS). SRTM DEM is useful to understand the location of the ponds based on elevation and slope. Fig 3A and 3B show the slope and elevation map of the study area. Fig 3A reveals that most of the study area is covered with a low slope, so it becomes more likely that surface water will be present at a location. A high value of slope will have less possibility of aquaculture area. Most of the aquaculture area lies in low elevation and low slope value. Ottinger et al. [23] also reported elevation and slope's role in identifying aquaculture areas in the low-lying coastal regions.

## 3.2. Secondary data and field survey

Secondary data were collected from district offices during the field visit in December, 2019. Following secondary data were collected from various government agencies. This include (i) aquaculture cover area, aquaculture production and export data collected from the District

Fisheries Office (DFO) in Satkhira, Bangladesh; (ii) rice production information from 2010 to 2019 was collected from Bangladesh Rice Research Institute (BRRI), Satkhira, Bangladesh.

The questionnaire survey and focused group discussion were also conducted with local people from 07[th] December to 26[th] December 2019 to understand the current situation about aquaculture, challenges, socio-economic condition of local communities, and local perspective and a possible solution. The survey and interview were done on a voluntary basis and no ethical permission was required before conducting this survey. During the questionnaire survey and focused group discussion, a structured interview and open-ended questions were used to gauge the socio-economic conditions and their views on the current and future status of aquaculture practices. We have asked questions to the head of the family, who is closely involved in aquaculture practices. In general, the average family size is six persons with low to medium monthly income. The questionnaire survey transcript is provided as supplementary file (see S2 File). S1 Fig shows the field photographs captured during the field visit to the study area.

### 3.3. Remote sensing data processing

**3.3.1. Sentinel-2 based NDWI and MNDWI processing.** Two commonly used water-indexing methods, i.e., NDWI and MNDWI were used to extract water bodies from Seninel-2 (S2) data based on literature. GEE platform and coding was used to generate NDWI and MNDWI indices from S2 surface reflectance images (see data availability section for the NDWI and MNDWI extraction codes). Table 2 shows an overview of S2 based NDWI and MNDWI. NDWI is useful for monitoring areas covered with water bodies but not suitable for build-up land and sporadically, overestimation occurred in water bodies [25]. NDWI was used to monitor open water bodies in this study [30]. The following equation was used to calculate NDWI.

$$NDWI = (GREEN-NIR)/(GREEN + NIR) \qquad (1)$$

where, Green (G) = 'B3'; Near-Infrared (NIR) = 'B8'

MNDWI is suitable for monitoring and mapping the water bodies as compared to NDWI [29, 30]. Thus, MNDWI index was also used to see the temporal changes of the water bodies and making a difference between NDWI and MNDWI indices. The threshold value of 0.04 was used for images from October to March (winter and pre-monsoon season) and 0.02 from April to September (summer and monsoon season) for aquaculture bodies extraction. Sentinel-2 based MNDWI was calculated using the following equation:

$$MNDWI = (GREEN - SWIR)/(GREEN + SWIR) \qquad (2)$$

where, Short Wave Infrared, (SWIR) = B11; Green, (GRN) = B3

**3.3.2. Digitization of aquaculture reference samples.** The aquaculture ponds are of different shapes and sizes in the study area. The object-based image classification algorithm is

**Table 2. Overview of Sentinel-2 based NDWI and MNDWI parameters.**

|  | Sentinel-2 (NDWI) | Sentinel-2 (MNDWI) |
|---|---|---|
| **Bands** | B3 (Green), B8 (NIR) | B3 (Green), B11 (SWIR) |
| **Formula** | NDWI = *(GREEN-NIR)/(GREEN +NIR)* | MNDWI = *(GREEN—SWIR) / (GREEN + SWIR)* |
| **Threshold Value** | -0.0607 | 0.04, 0.02 |
| **Data Processing** | Google Earth Engine | Google Earth Engine |
| **Image Acquisition Period** | 2017, 2018, 2019 (3 years) | 2017, 2018, 2019 (3 years) |

useful for identifying the shape of aquaculture ponds. A total of 4,672 aquaculture ponds were visually digitized using high-resolution Google Earth images. These digitized aquaculture pond data were used as sample data for further analysis. Although Google Earth provides high-resolution satellite data, it is not available for the entire study area. We have considered two points, while digitizing sample aquaculture ponds data using Google Earth, that is, a) the acquisition date should not be older than January 2018, and (ii) only permanent aquaculture ponds should be mapped where paddy was not cultivated. To extract qualitative data from the sample aquaculture ponds in the study area, the perimeter and area of the digitized ponds were calculated using Eq 3. Furthermore, compactness metrics were analyzed to calculate the complexity of the shape of the aquaculture pond using Eq 4

$$P2A = \text{perimeter}^2/\text{area} \tag{3}$$

$$\text{Compactness } C = \sqrt{\text{Area}}/Bi, \tag{4}$$

where, $Bi = \text{perimeter}^2/4\pi$ (Ottinger et al. [14])

The compactness index is dimensionless, which means that the scale of the object was not influenced, and it has a value of 1 for a circle and a spectrum of 0 to 1 for all shapes of the plane. The meaning differs between various shape metrics, but higher values generally indicate greater complexity of the shape [31]. Fig 4 shows the shape statistics of digitized aquaculture ponds structure in the study area. The Kaligange area aquaculture ponds' shape and compactness are larger as compared to other areas. For each shape metric we used, the non-parametric Mann Whitney U test (also known as Wilcoxon rank sum test) was conducted to see any significant difference in the metric values between the 5 sites. The analysis was done in R language v. 4.1.3 with the 'wilcox_test()' function in the package 'rstatix'. The test was conducted for each combination of sites, with a null hypothesis that there is no shift in the distribution of site 1 and 2, and alternative hypothesis is that group 1 is shifted to the left of group 2. We noticed that for the area metrics, Satkhira Sadar had significantly larger ponds than Assassuni ($p < 0.001$), Kaligange ($p = 0.009$) and Debhata ($p = 0.026$). However, Satkhira Sadar's pond perimeter was only significantly larger than those of Assassuni ($p = 0.039$). For the compactness metrics, Debhata had significantly higher values than Assassuni ($p < 0.001$) and

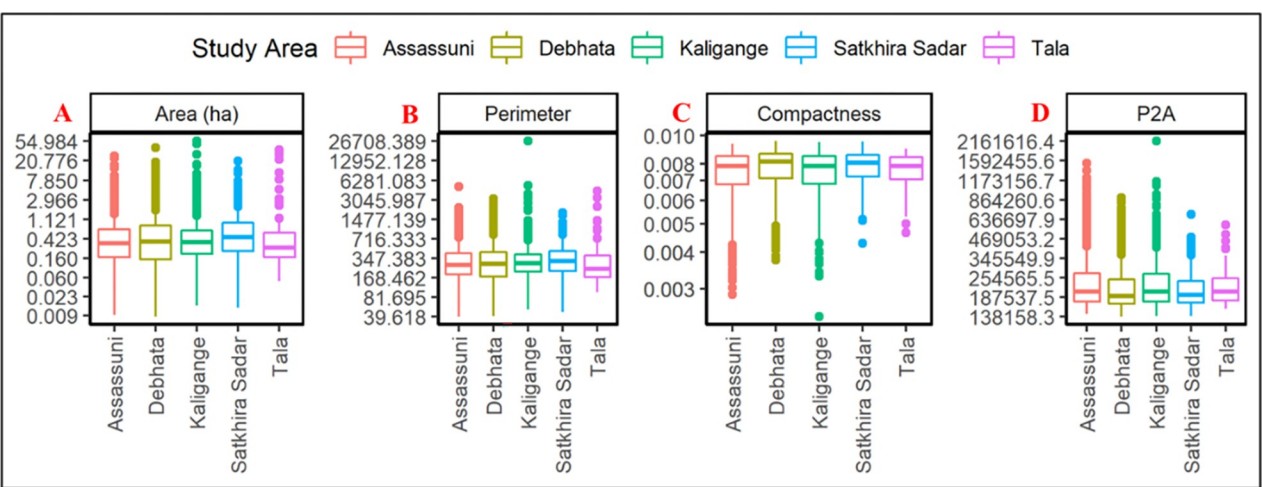

**Fig 4.** Box plots shows the shape metrics calculation: (A-Area (ha), B-Perimeter, C-Compactness, and D-P2A) for the study area aquaculture ponds samples.

Kaligange (p <0.001), and similarly Satkhira Sadar had significantly higher values than Assassuni (p = 0.006) and Kaligange (p = 0.008). For the P2A metric, Kaligange was higher than Debhata (p < 0.001) and Satkhira Sadar (p = 0.08), and Assassuni was also higher than Debhata (p < 0.001) and Satkhira Sadar (p = 0.06). All reported p-values are adjusted for multiple comparisons using the Holm method.

**3.3.3. Multi-temporal Sentinel-1 data processing.** Sentinel-1 images were used for segmentation and object-based image classification in this study. The following three steps were used to extract the objects using sentinel-1 data, i.e., a) image thresholding, b) water mask and segmentation, and c) object-based classification. Du et al. [32] reported the importance of threshold selection for water body mapping on the Venice coastland, Italy. Threshold values vary spatially and temporally, depending on the backscattering information in the image. Various automatic threshold selection methods are available to distinguish water and other objects. In this study, we used the Iterative Self-Organizing Data Analysis Technique (ISODATA) and Otsu thresholding based on previous studies. ISODATA thresholding is a means of automatically finding a threshold for a given grey image value, where the mean pixel values in the two categories are generated (objects and background) by applying a binary threshold value [33]. This method is important for image analysis and pattern classification. On the other hand, Otsu thresholding is a process that can find the optimal threshold to adaptively differentiate two-class data [34]. This technique can be used for image segmentation and binarization based on histogram shape [35]. The algorithm is widely used to maximize variance between classes and minimize variance intraclass [35]. Otsu automatically defines a threshold value $t$ that divides the image into water and non-water classes. The value of $t$ is determined by the following equations:

$$\delta^2 = P_{nw}.(M_{nw}-M)^2 + P_w.(M_w-M)^2 \tag{5}$$

$$M = P_{nw}.M_{nw} + P_w.M_w \tag{6}$$

$$P_{nw} + P_w = 1 \tag{7}$$

$$t = \underset{x \le t \le y}{\text{Arg Max}} \ \{P_{nw}.(M_{nw}-M)^2 + P_w.(M_w-M)^2\} \tag{8}$$

where, $\delta$ is the inter-class variance of the non-water class and water class; $P_{nw}$ and $P_w$ are the probabilities of one pixel belonging to non-water and water, respectively; $M_{nw}$ and $M_w$ are the mean values of the non-water and water classes; and $M$ is the mean value of the feature image.

Fig 5 shows the histogram of the calculated temporal median image based on Sentinel-1 data with VV (vertical transmitting, vertical receiving) and VH (vertical transmitting, horizontal receiving) polarization. The range of backscattering coefficient in VV polarization is wider as compared to that in VH polarization. In this study, we tested the suitability of VV and VH polarization to detect water bodies and aquaculture areas. Fig 5A and 5B show the histograms of the ISODATA-VV and VH polarization mode to detect the presence of the water bodies. Fig 5C and 5D show the histograms of Otsu-VV and VH polarization mode to detect water bodies. The Otsu-VH histogram-based threshold shows two distinct peaks, so it is suitable for the separation of bimodal distributions [36]. Otsu-VH polarization-based threshold shows a better result for detecting water bodies than other thresholds with different polarization. Therefore, Otsu-VH polarization-based threshold was applied for the separation of water bodies. Ottinger et al. [14] also reported the use of Otsu-VH polarization to identify water cover areas in Vietnam.

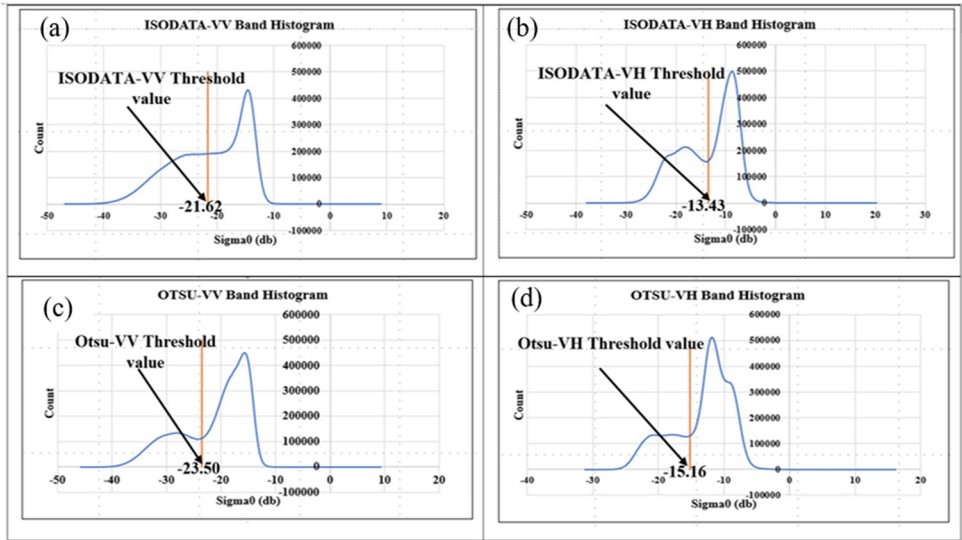

**Fig 5.** Histograms of (a) ISODATA-VV, (b) ISODATA-VH, (c) Otsu-VV and (d) Otsu-VH bands.

After applying the thresholds, the segmentation and classification of images was performed in the Orfeo toolbox. This toolbox is helpful in detecting an object from the land and a convenient and fast way to apply different parameters [37, 38]. This can optimize to extract the appropriate parameters to detect different objects. In this study, a multi-resolution segmentation algorithm was applied to the Sentinel-1 Otsu-VH water mask treated images. We have also used slope and elevation data to mask out other classes before performing the object-based classification. In the study area, there are three types of classes identified based on field-work, *viz.* (i) only aquaculture, (ii) paddy and (iii) paddy with fish farming. Farmers did not use the same land all the time to cultivate paddy, or aquaculture and paddy with fish farming. Based on the farmers' requirement and the availability of water, they keep changing the size of the aquaculture area, paddy and paddy with fish farming. There would be a high possibility of misclassification if a simple image-based classification algorithm was applied to classify satellite data. To solve the misclassification problem, we used a masked image for classification. Bare lands, roads, rivers, built-up areas and orchard was masked before applying object-based classification.

**3.3.4. Validation.** The Accuracy Assessment was conducted to evaluate the classification performance. The most common method to assess the accuracy of a classified map is to generate a series of random points and use these random points in a confusion matrix based on ground truth data and correlate with the classified data [39]. Comparing the outcomes of various classification strategies or training sites is important. Google Earth Imagery and ground-truth data were used as reference data in this study and compared with the classification results. To assess the accuracy of the aquaculture mapping, a confusion matrix was calculated for each image. Producer accuracy (PA), user accuracy (UA), and overall accuracy (OA) was generated using the following equations [40]:

$$UA = \frac{(Number\ of\ correctly\ classified\ pixels\ in\ each\ class)}{(Total\ bumber\ of\ classified\ pixels\ in\ that\ class)}\ x\ 100$$

$$PA = \frac{(Number\ of\ correctly\ classified\ pixels\ in\ each\ class)}{(Total\ bumber\ of\ reference\ pixels\ in\ that\ class)}\ x\ 100$$

$$OA = \frac{(Total\ number\ of\ correctly\ classified\ pixels)}{(Total\ bumber\ of\ reference\ pixels)}\ x\ 100$$

## 4. Results

### 4.1. Sentinel-2 (NDWI, MNDWI) based aquaculture water surface

This study explored the use of NDWI and MNDWI methods to extract water bodies based on the previous studies [30, 32]. OTSU thresholding was used to extract water bodies to test the extraction of aquaculture water surfaces. The threshold values for NDWI and MNDWI were calculated for every image based on the OTSU method (Table 3) to extract. These thresholds were applied to cloudless images to find the temporal changes of the water bodies. Fig 6 shows multitemporal changes in the water bodies covered area using the NDWI threshold. The NDWI can extract most of the water bodies, including muddy areas. The month of March shows the lowest value in water bodies compared to December or January in the same year. This is due to the dry period in the month of March. Fig 7 shows the multitemporal water body extraction based on the MNDWI threshold. The information of MNDWI based water bodies also shows the lowest value in water bodies in March. Fig 8 shows the temporal variations of the water bodies based on NDWI and MNDWI. The results show that the water bodies extracted by NDWI show an overestimation compared to MNDWI (Fig 8). In this study, MNDWI performs better in the extraction of water bodies compared to NDWI. The extraction of aquaculture needs multi-temporal data because most of the aquaculture ponds are filled with water all year round and partially drained during the time of harvesting. Singh et al. [30] also reported that MNDWI performs better than NDWI in extracting water bodies mixed with vegetation.

### 4.2 Sentinel-1 (SAR) based aquaculture area

Taking into account the Sentinel-1 backscatter information and the existing knowledge of the study area, three classes of land use were identified and classified, respectively, (a) aquaculture pond, b) paddy field and c) paddy with fish farming. In Satkhira, there are three cycles of paddy cultivation with different varieties of rice each year. Table 4 shows the seasonal rice and shrimp cultivation cycle in different seasons in the study area. The rice cultivation time varies

**Table 3.  Otsu's thresholding for NDWI and MNDWI.**

| Date | NDWI Otsu's thresholding | MNDWI Otsu's thresholding |
|---|---|---|
| Jan, 2017 | 0.02 | 0.44 |
| Jan, 2018 | 0.02 | 0.43 |
| Jan, 2019 | 0.02 | 0.41 |
| Feb, 2017 | 0.02 | 0.37 |
| Feb, 2018 | 0.03 | 0.37 |
| Feb, 2019 | 0.02 | 0.40 |
| Mar, 2017 | 0.05 | 0.51 |
| Mar, 2018 | 0.02 | 0.45 |
| Mar, 2019 | 0.04 | 0.46 |
| Dec, 2017 | 0.07 | 0.54 |
| Dec, 2018 | 0.04 | 0.51 |
| Dec, 2019 | 0.04 | 0.46 |

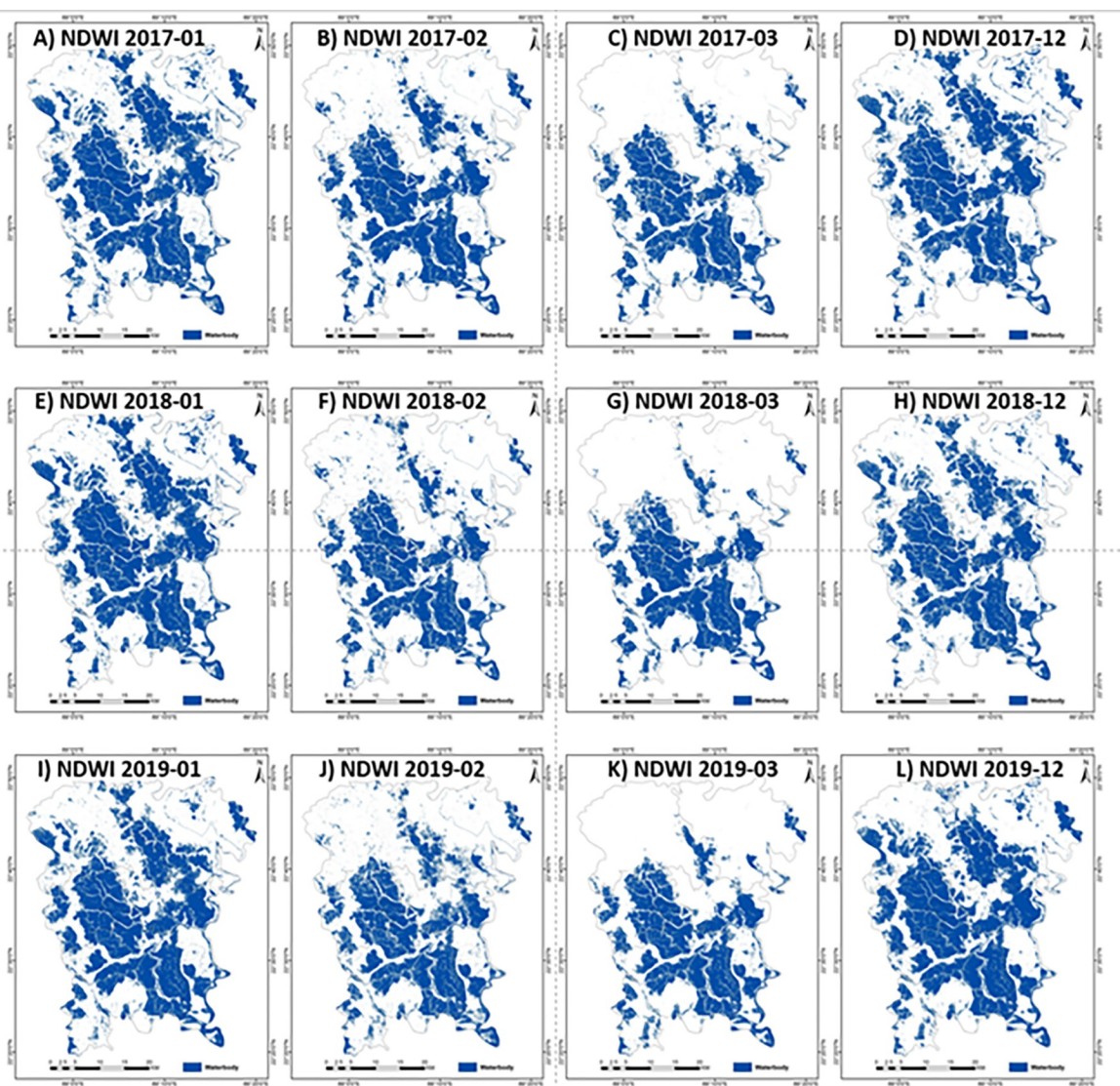

**Fig 6.** Multi-temporal Sentinel-2 based NDWI images of (A) January-2017 (B) February-2017 (C) March-2017 (D) December-2017 (E) January-2018 (F) February-2018, (G) March-2018 (H) December-2018, (I) January-2019 (J) February-2019, (K) March-2019, and (L) December-2019.

with the rice type and cultivation practices, e.g., some farmers cultivate only rice, some farmers cultivate rice and fish in the same field, and some farmers do aquaculture for fish production. In addition, they do aquaculture in the rest of the seasons or months. So, there were some drastic changes observed in the land cover pattern in different months of the year. Furthermore, the water volume was high during the rice sowing season, which was difficult to differentiate between paddy fields, aquaculture areas, and paddy with fish farming using only satellite data. In this study, the rice harvesting time was selected to understand the difference between paddy fields, aquaculture areas, and paddy fields with fish farming. The seasonal calendar of shrimp farming shows December as the showing time, January to March as the growth stage, and April/May as the harvest period. This season varies with the location and availability of saline/fresh water and shrimp species. Fig 9 shows the object-based image classification results using multi-temporal Sentinel-1 data.

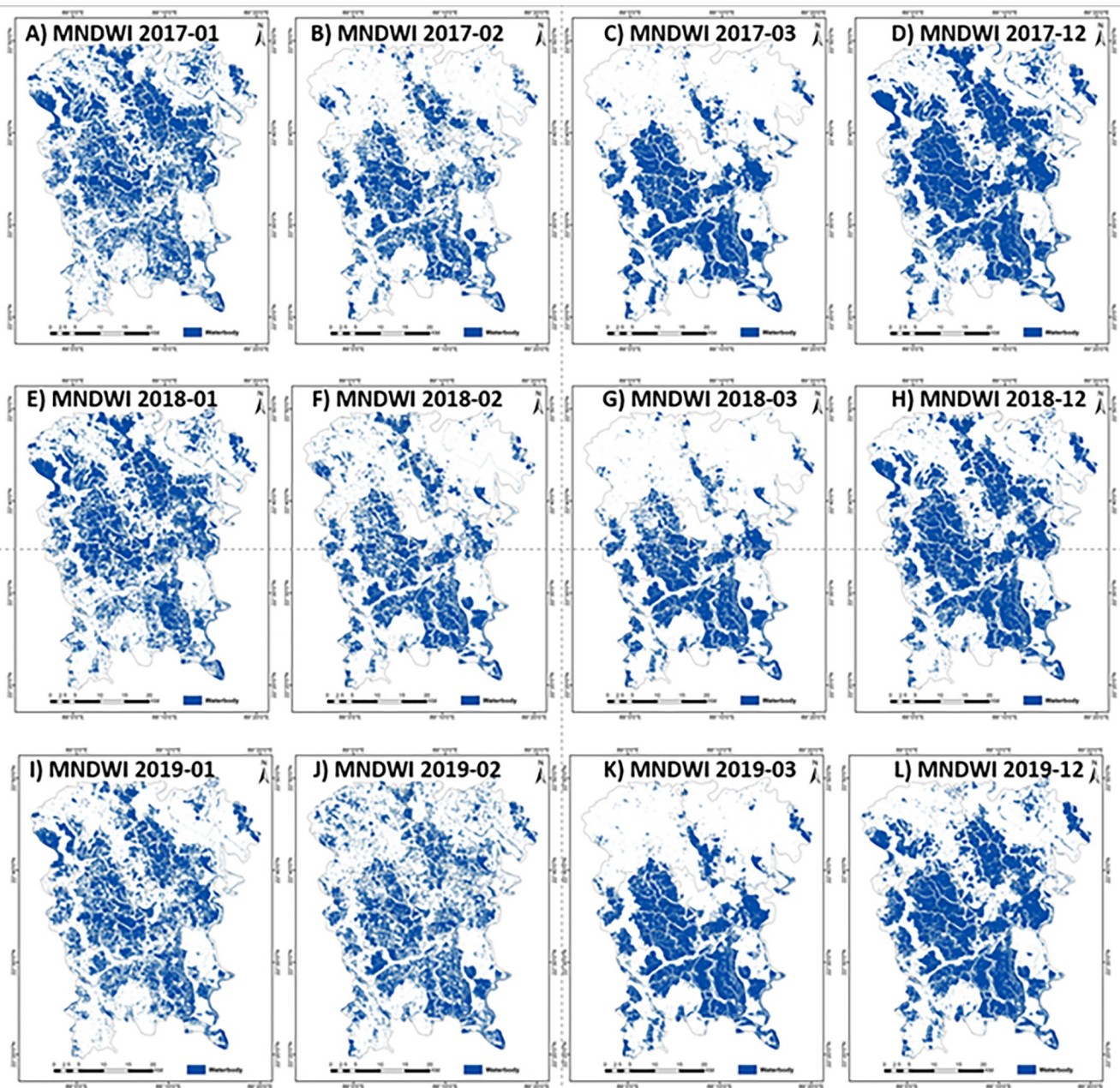

**Fig 7.** Multi-temporal Sentinel-2 based MNDWI images of (A) January-2017 (B) February-2017 (C) March-2017 (D) December-2017 (E) January-2018 (F) February-2018, (G) March-2018 (H) December-2018, (I) January-2019 (J) February-2019, (K) March-2019, and (L) December-2019.

Fig 9 shows the temporal changes of three classes, *viz.* aquaculture area, paddy fields and paddy with fish farming, from 2017, 2018 and 2019. For the months of January and December, most of the study area was covered with aquaculture, followed by pastures and pastures with fish farming. The paddy cultivation period is not consistent in the study area due to the different varieties of rice. However, in May and August the field of the paddy was comparatively more dominant than the other two classes. This is because the harvesting season for Boro and Aus rice is between May and August, and during this season, the water level becomes lower because of the summer season.

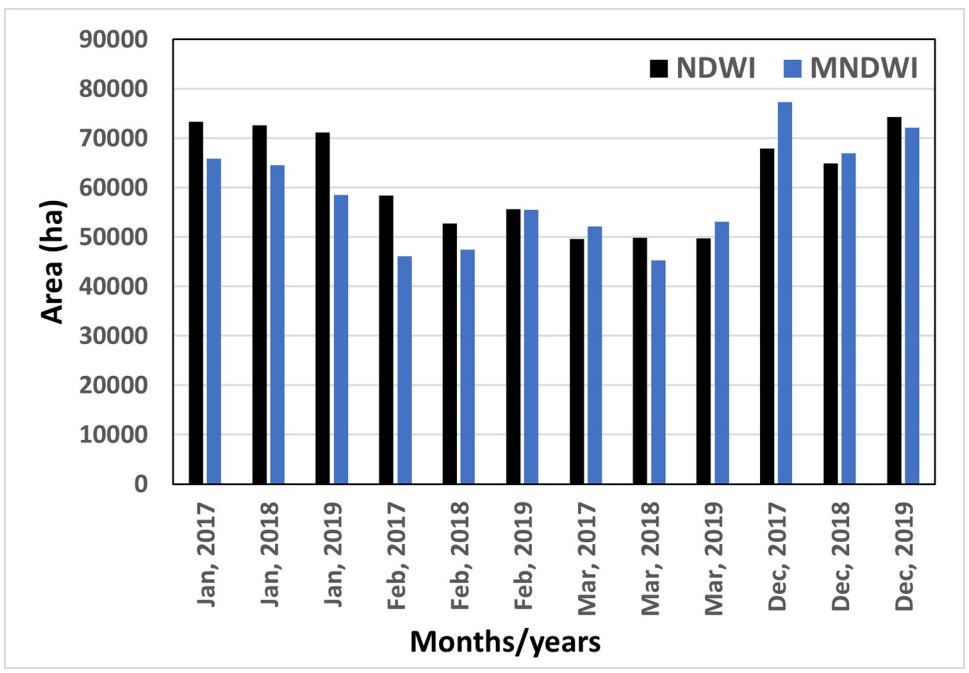

**Fig 8. Temporal variations of water bodies based on Sentinel-2 NDWI and MNDWI in the study area.**

Fig 10 shows the multi-temporal changes in aquaculture, paddy fields, and paddy with the fish farming classification results of Sentinel-1 data from 2017 to 2019. During the months of January and December for the years between 2017 and 2019, the extent of the aquaculture area was high except for January 2018. On the other hand, the paddy area was high between May and August as compared to other months. However, only a smaller areal extent is observed for the class paddy with the fish farming area, and its annual change is also significantly low compared to other classes.

We calculated the accuracies of the classified maps based on Sentinel-1 data from 2017–2019, using producer's accuracy, user's accuracy, and overall accuracy. Training points for accuracy assessment were used from the field survey as well as Google Earth Images. S2 Table shows the confusion matrix to assess the accuracy of the classification. The overall accuracy was above 80%, indicating a good model output.

### 4.3. Secondary data and field survey

To validate remote sensing results, we had also collected secondary data and a questionnaire survey and focused group discussion in December 2019. We have also visited various

**Table 4. Rice cultivation cycle in different seasons.**

| Rice Type | Jan | Feb | Mar/Mid Mar | Apr/Mid Apr | May/Mid May | Jun | Jul | Aug/Mid Aug | Sep | Oct | Nov/Mid Nov | Dec |
|---|---|---|---|---|---|---|---|---|---|---|---|---|
| Boro Rice | ▨ | | | ▨ | ▨ | | | | | | ▨ | ▨ |
| Aus Rice | | | ▨ | ▨ | ▨ | | | ▨ | | | | |
| Aman Rice | | | | | ▨ | ▨ | ▨ | ▨ | | | ▨ | ▨ |
| Season | Winter | | Summer/Pre-monsoon | | | | | Monsoon | | Post-Monsoon | Winter | |

Yellow: Sowing

Green: Harvesting

Source: (Nelson et al., [13])

**Fig 9.** Sentinel-1 object-based classification results: (A) Jan-2017, (B) May-2017, (C) Aug.-2017, (D) Dec.-2017, (E) Jan.-2018, (F) May-2018, (G), Aug.-2018, (H) Dec.-2018, (I) January 2019, (J) May-2019, (K) August 2019 and (L) December 2019.

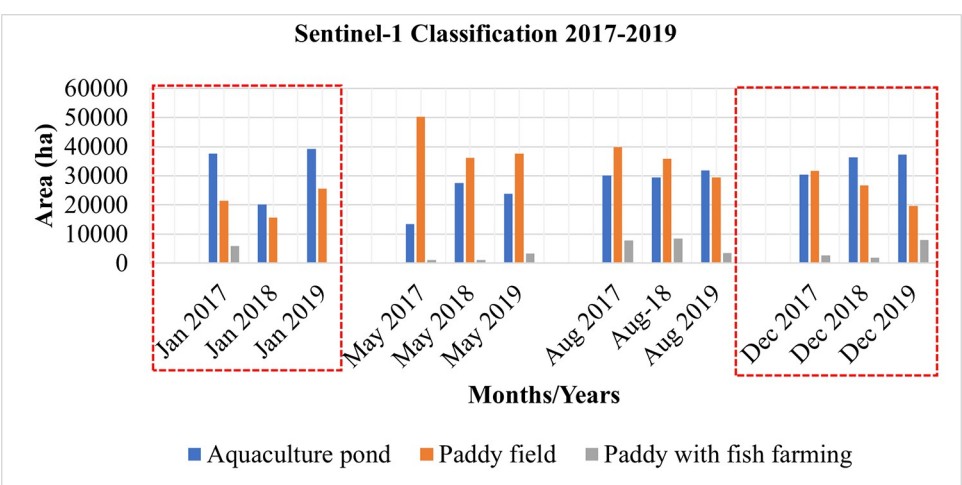

**Fig 10. Sentinel-1 object-based image classification result (2017, 2018, 2019).**

government offices to gather relevant information like crop harvesting, export, challenges etc. The key findings of the above exercise regarding aquaculture is presented below.

**4.3.1. Temporal change in shrimp farming and rice production.**   Generally, *Penaeus monodon* shrimp (locally called Bagda) are cultivated mainly in the Kaligange, Assassuni and Debhata areas where brackish water is present. On the other hand, the freshwater prawn *Macrobrachium Rosenbergii* (locally called Galda) is mostly limited to coastal areas, but it is slowly cultivated and expands in the field of development. Galda can also be grown in conjunction with paddy cultivation in Satkhira and Tala. In the Kaligange, Assassuni, and Debhata areas, the soil and groundwater are saline, which is very suitable for Bagda farming. Fig 11 shows the production of Bagda (*Penaeus Monodon*) and Galda (*Macrobrachium Rosenbergii*) in the study area. Bagda production is high in Assassuni, Kaligange, and Debhata as compared to the Satkhira and Tala areas because of the availability of brackish water.

Rice production data was collected from the Bangladesh Rice Research Institute from 2010–2019. Fig 12 illustrates the rice production data in various subdistricts of the Satkhira district from 2010–2019. Rice production in Satkhira and Tala was higher than in Debhata, Kaligange and Assassuni areas due to freshwater availability. In Satkhira, aquaculture induced a dramatic change in the livelihoods of the coastal poor, especially for women [41]. Although farmers in this region predominantly cultivate rice, a recent shift toward aquaculture production is observed mainly because of the financial benefits. Fig 12 shows the decreasing pattern of rice production in the Satkhira district from 2010–2019 [42]. On the other hand, there has been a drastic increase in shrimp farming in Bangladesh and Satkhira for the last two decades, providing tremendous economic benefit and livelihood to local farmers. Furthermore, the country has gained significant international revenue from shrimp production and export [43].

Fig 13 shows the shrimp production and export data from shrimp farming from 2010 to 2018 in Satkhira collected from the District Fisheries Office. Shrimp production is observed to show an increasing trend from 2010 to 2013 and there is not much change from 2014 to 2019. Shrimp export shows a decreasing trend from 2011 to 2011. There are several causes such as: (a) Inability to maintain shrimp quality or failure to abide by the international law on food health and safety, (b) lack of infrastructure maintenance, (c) Unable to keep up with

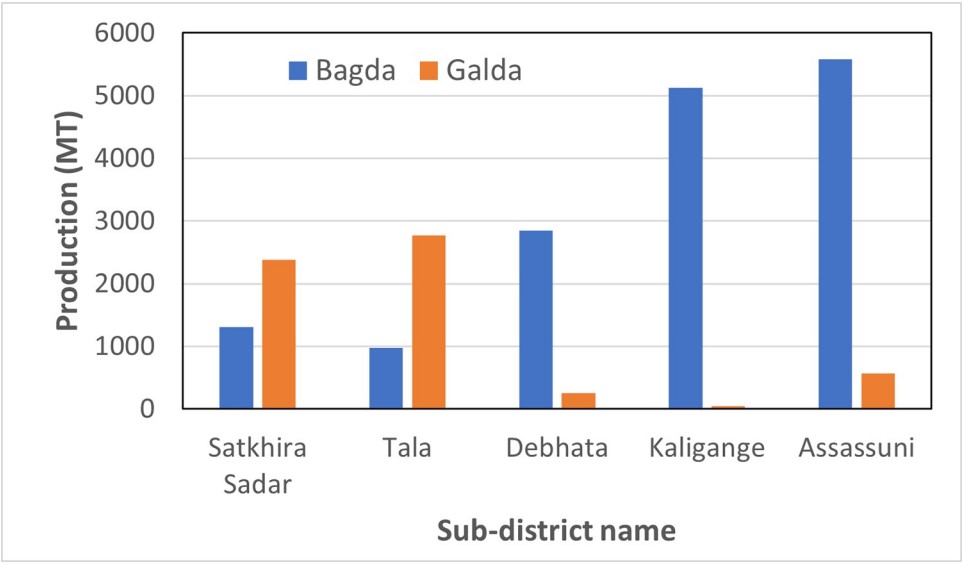

**Fig 11.  Bagda (*Penaeus Monodon*) and Galda (*Macrobrachium Rosenbergii*) production in the study area [25].**

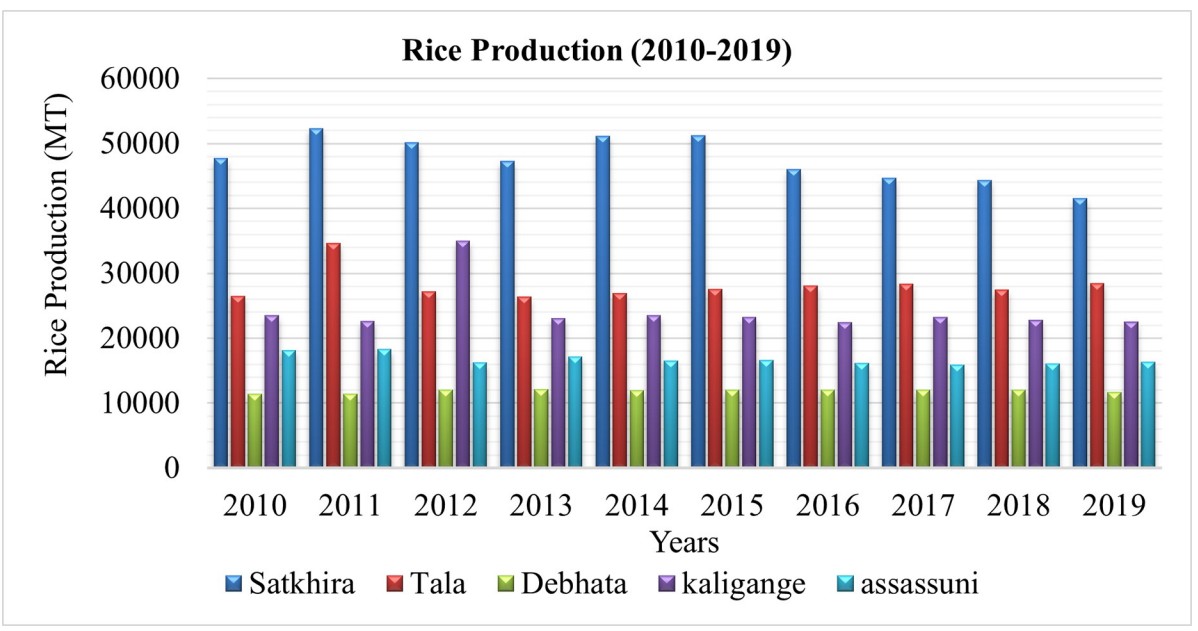

**Fig 12. Rice production in the study area in the last ten years.** [Source: Bangladesh Rice Research Institute (BRRI)].

international market value, (d) political issues, (e) outbreak of diseases, (f) natural calamities like cyclone, flood etc., and (g) Overuse of naturally available nutrients, that is, cultivating shrimp continuously for 15 to 25 years will decrease the production.

**4.3.2. Local people involvement in aquaculture.** Although Bangladesh is an agricultural country, farmers are interested in fish farming because of its monetary benefits. Most of the local people are involved in aquaculture cultivation. Fig 14 shows the percentage of people involved in aquaculture and rice and other crops cultivation in different subdistricts based on the questionnaire survey data. Fig 14A shows that most of the respondents in Assassuni (32%)

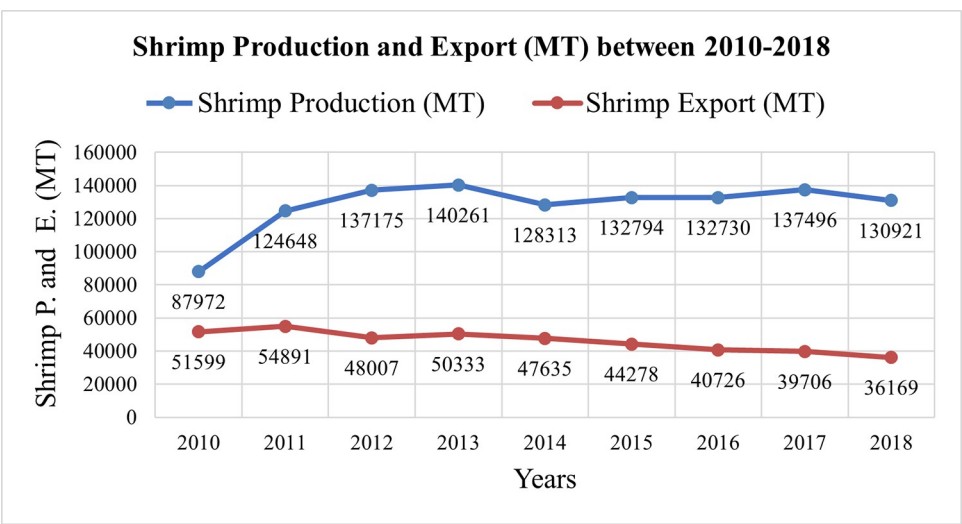

**Fig 13. The production and export data of shrimp farming from 2010 to 2018 years.** [Data Source: District Fisheries Office (DFO), Satkhira] [Note: P = Production and E = Export].

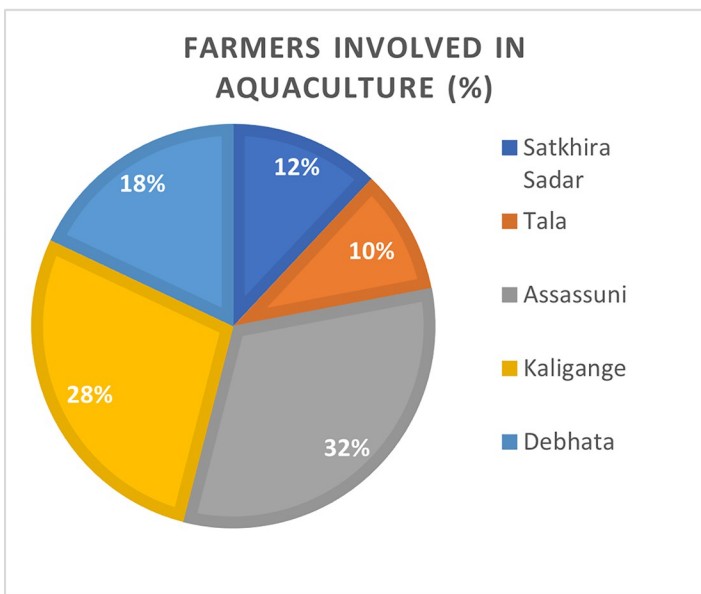
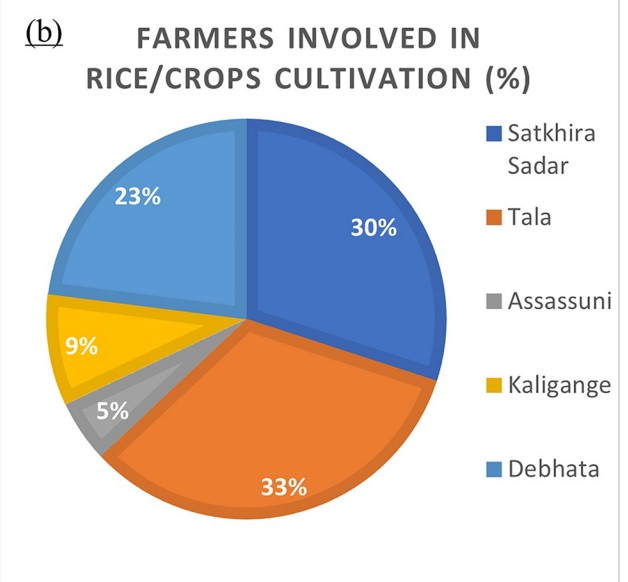

**Fig 14.** Percentage of farmers involved in (a) aquaculture, (b) rice/crops cultivation in the study area. [Data source: questionnaire survey].

and Kaligange (28%) subdistricts are involved in aquaculture. However, the respondents in Tala (33%) and Satkhira (30%) are involved in rice / crop cultivation (Fig 14B). This is due to the presence of fresh water in the Tala and Satkhira subdistricts.

**4.3.3. Impacts of shrimp cultivation.** The extension of shrimp production has modified the trend of land use and had a detrimental influence on the coastal habitats of Bangladesh, as well as Satkhira [44, 45]. Common environmental consequences from aquaculture include mangrove destruction, saltwater intrusion, sedimentation, waterlogging, and pollution, and outbreaks of diseases. On the other hand, shrimp production also negatively impacts rice, crops, and vegetables such as coconut, mango, giant taro, jackfruit, and blackberry production [46].

**4.3.4. Strategies for shrimp farming in harmony with the environment.** Despite its favorable environment, shrimp production in Satkhira is under severe environmental threat. Therefore, government departments and NGOs may develop a framework including certain strategies for sustainable shrimp farming, as shown in Fig 15. This would enable farmers to improve their production and help to save the environment by maintaining the balance between aquaculture, traditional crop production, and livestock rearing.

a. Cluster farming: Out of several approaches, cluster farming can be adopted to witness a significant increase in shrimp production [45]. Farmers have the advantage of obtaining systematic training to learn about advanced technologies for sustainable agricultural practices through cluster farming. Also, it is much easier to get loans or other benefits from the financial institutions for the farmer group organized under cluster farming. Globally, including some Asian countries such as Vietnam, Indonesia, the Philippines, and India, cluster farming has proven successful [46, 47].

b. Cultivation of salinity-tolerant plants: Since land and water have already become saline in several pockets of the study areas, farmers may opt to use salinity-tolerant plants more often in their fields. The most common types of salinity-resistant crops and vegetables are

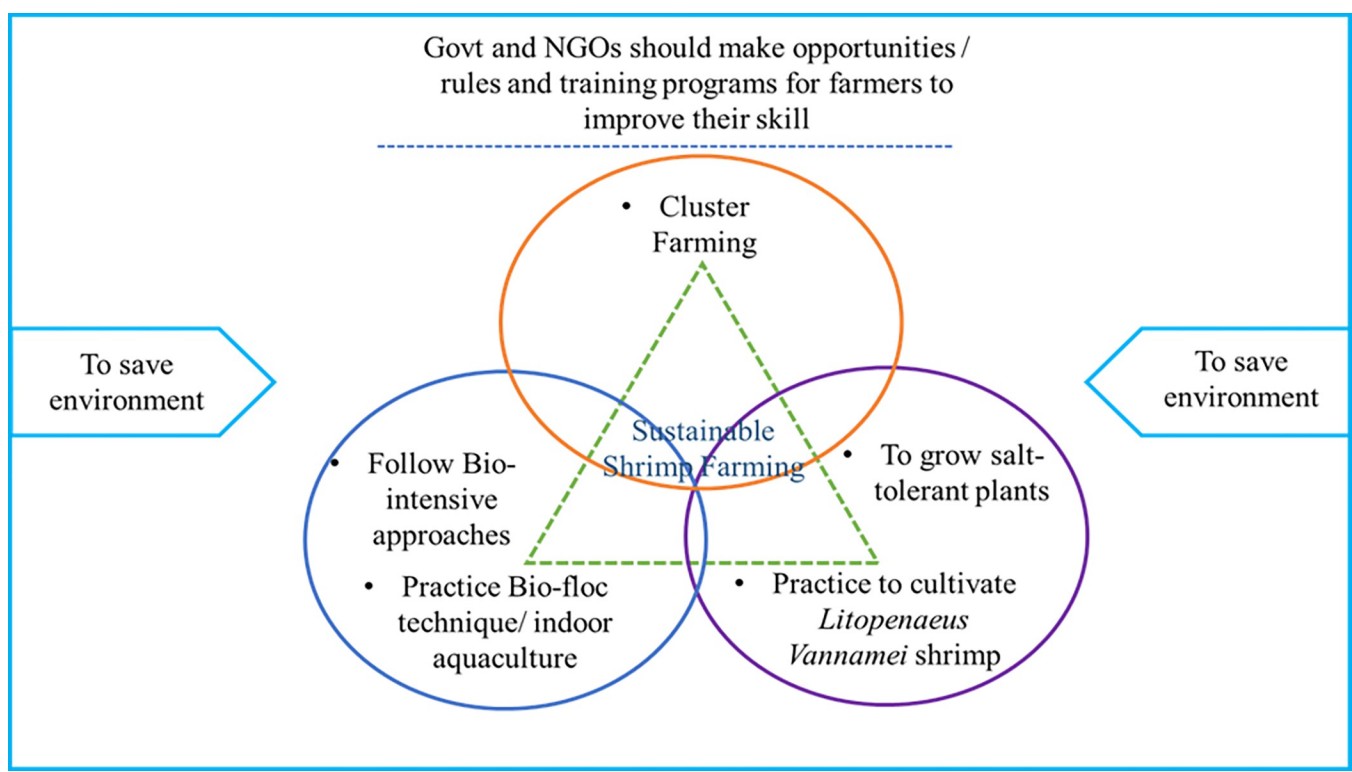

**Fig 15. Strategies for sustainable shrimp farming and to save the environment.**

Date palm, Cabbage, Coconut palm, Sugar beet, Garden beet, Lettuce, Carrot, Spinach, Potato, Tomatoes, Sweet potato, Asparagus etc. This kind of practice would be beneficial both in the environmental and economic aspects [48].

c. Practice bio-intensive approaches: Shrimp cultivation is significant for financial benefits, but it also has some negative environmental effects. As a countermeasure, if farmers cultivate shrimp every alternate year by cultivating different vegetables and crops in the gap year, that can help the soil to regain nutrients and maintain ecosystem balance.

d. Cultivation of hybrid shrimp species: Bagda shrimp (*Penaeus monodon*) is cultivated in brackish water with high salinity tolerant capacity and cultivation of Galda shrimp (*Macrobrachium Rosenbergii*) in freshwater with low salinity tolerant capacity. Now, farmers are growing *Litopenaeus Vannamei* shrimp, which can be grown in freshwater and brackish water [49]. It has low environmental impact, which could be helpful in reducing the negative impact of brackish water Bagda (*Penaeus monodon*) shrimp as well as monetary benefits.

e. The practice of Biofloc technique: Nowadays, indoor shrimp culture or Bio-floc technique is in practice in many countries like USA, Indonesia, Singapore, Malaysia etc. This technology extracts inorganic nitrogen from wastewater from aquaculture and enhances water quality by balancing nitrogen and carbon [50, 51]. It is beneficial for the environment, enhances the shrimps' taste value resulting in more profit. In Satkhira, some private firms are also using Biofloc/ indoor shrimp aquaculture to increase shrimp cultivation but still need promotional activities to create awareness among small-scale farmers.

## 5. Discussion

This study focused on mapping aquaculture areas in the Satkhira region. Multisensor remote sensing data were used to monitor aquaculture areas. Sentinel-2-based NDWI and MNDWI were used to analyze temporal changes in water bodies. The result shows that NDWI overestimated the extent of water bodies, whereas MNDWI is more suitable for detecting water bodies, which is similar to some of the previous studies [29, 30, 32]. Sentinel-1 (SAR) images using OTSU-VV and VH and ISODATA binary threshold values were used to identify water bodies and understand which mode of polarization is appropriate for extracting the water bodies. The results show that OTSU-VH is more suitable for detecting water bodies as compared to VV polarization. Object-based image classification was used to monitor the temporal changes and spatial extent of the aquaculture area. In Satkhira, there are three cycles of paddy cultivation with different varieties of rice each year. Some farmers cultivate only rice, some farmers cultivate rice and fish in the same field, and some farmers do only aquaculture. Therefore, some significant changes in the land cover pattern have been observed in different months of the year. Monitoring only aquaculture areas was challenging due to the varied use of land throughout the year. In this study, we classified the study area into aquaculture, pasture field and pasture with fish farming. The general classification accuracies were above 80%.

Previous studies also applied object-based image analysis methods to identify aquaculture areas [14–18]. They have applied a connected component segmentation method using Sentinel-1 with very high-resolution satellites such as SPOT-5, WorldView-1, and Pleiades along with different indexes such as NDWI, MNDWI, AWEI in their research. In addition, they applied edge sharpening to obtain the appropriate shape of the aquaculture pond. Satkhira is one of the main aquaculture hotspots in Bangladesh, where shrimp aquaculture is reported to be practiced in the vicinity of paddy fields. Due to this, an increase in soil salinity is observed, consequently reducing profit from paddy cultivation. Shahbaz et al. [48] reported that a 10% increase in shrimp farm-induced salinity reduces paddy farm profits by 1% to 3% in a subdistrict of Satkhira district. However, the intraannual temporal utility of such extracted aquaculture ponds was not discussed in these studies. Furthermore, the present condition of the ponds and their implications on the surrounding environmental impact were also not discussed. Taking these into account, in our research, we tried to adopt the following methods to fulfill the following research gaps.

a.  Multi-resolution segmentation algorithm was used in this study.

b.  In the Object-based image classification algorithm, other land cover types were masked to avoid misclassification.

c.  Three classes were monitored. For example, aquaculture area, paddy field and aquaculture paddy field, because in different months, the farmers practice either aquaculture or paddy or paddy with fish farming.

d.  Secondary data and field survey data were used to understand the current perspective and the real-world situation of the study area and validation of this study.

During the design of different adaptation and mitigation measures (applying new technologies for water access, shrimp food formulation, good storage system, etc.) for sustainable aquaculture, it is important to consider land-poor and capital-poor farmer communities, as they account for a large percentage of people involved in aquaculture in developing nations [52]. Sustainable intensification in terms of shifting the focus from volume to value is the key to maintaining a dynamic equilibrium of the global value chain (from producer to consumer) and sustainable food production, which is highly relevant for global food security [53]. To

minimize the negative impacts of aquaculture on the environment and provide better credibility for market prices and consumer perceptions, environmental certification is essential. Also, the most common way to achieve this certificate is to follow the factors and their impact areas provided in the FAO Technical Guidelines on Aquaculture Certification for Responsible Shrimp Farming [54]. Brackish water aquaculture, including multitrophic aquaculture, is one of the most potential adaptation options for a region like Satkhira, where low-income poor farmers are in large numbers and a large portion of land and water became salinized [55]. With a look at the scarce water resources and the increasing food demand and energy sources, it is essential to look at both synergies and trade-offs. It can be dealt with holistically through the lens of the water-food-energy nexus rather than dealing with them in silos and looking at short-term economic incentives [56].

There are some limitations in this study that can be further addressed in future research. a) Image resolution is important to obtain high accuracy and detect small aquaculture ponds. A very high-resolution sensor, for instance, WorldView and Pleiades would improve the accuracy of aquaculture monitoring; however, these are commercial satellites. b) During our fieldwork, the water quality was not measured. Measurement of water quality can help us understand the level of salinity, water temperature, dissolved oxygen, contaminants, and any other chemical elements present in the water. This would substantially improve our understanding of the spatial disparity in shrimp production. c) Though farmers are well informed, they did not want to openly share information about the production, advantages, or disadvantages of aquaculture and how to solve related problems during the questionnaire survey. d) The lack of up-to-date government data (secondary data) hampers our output in a certain way.

## 6. Conclusion & recommendations

Aquaculture is crucial in regions that rely heavily on fish for food. Therefore, it is important to monitor spatio-temporal change in the characteristics of aquaculture. This study examined various approaches to monitor temporal changes in aquaculture using Sentinel-1-based radar backscattering intensity and Sentinel-2-based NDWI and MNDWI data in the Satkhira district, Bangladesh, between 2017 and 2019. Sentinel-2-based NDWI overestimated the extent of water bodies. However, Sentinel-2 based MNDWI is more suitable for detecting waterbodies accurately. OTSU-VH polarization is more suitable for detecting water bodies as compared to VV polarization. Sentinel-1 data were used to classify areas as aquaculture pond, paddy field or paddy field with fish farming using object-based classification. The results of the classification revealed that the extent of the aquaculture area was higher compared to paddy and paddy with fish farming areas, except during the dry season in May, while the extent of paddy fields and paddy with fish farming areas did not change significantly during the study period. Based on the questionnaire survey and secondary data, we noticed the presence of brackish water is suitable for the cultivation of brackish-water shrimp (*Penaeus monodon*) in Kaligange, Assassuni, and Debhata subdistricts. The presence of freshwater in the Satkhira, Sadar and Tala areas is suitable for the cultivation of *Macrobrachium rosenbergii*.

To improve the sustainability of aquaculture in the region, we recommend the promotion of better aquaculture and farming techniques. For instance, the Biofloc fish farming technique/Indoor aquaculture, Bio-intensive approaches, growing salt-tolerant crops and the cultivation of *Litopenaeus Vannamei* shrimp could be useful for preserving the ecosystem's balance and monetary benefits. Farmers should consider farming rice and shrimp together. Moreover, in order to improve agriculture in such saline-prone areas, it is also important to raise knowledge of modern cultivation techniques like proper soil management, fertilizer usage, crop

rearing and alternative cropping systems. In addition, extensive research should be undertaken to identify new methods for ensuring greater food security in saline-prone regions in Bangladesh. The finding of this study might be useful for achieving certain Sustainable development goals, primarily SDG 2 (zero hunger), SDG 6 (clean water and sanitation) and SDG 8 (decent work and economic growth) and SDGs 12 (sustainable consumption).

## Supporting information

**S1 Table. List of Sentinel-2 data acquisition.**
(PDF)

**S2 Table. Accuracy assessment of the sentinel-1 (SAR) classification.**
(PDF)

**S1 File. GEE code to generate MNDWI data and maps.**
(PDF)

**S2 File. Questionnaire survey for aquaculture and shrimp farming Satkhira, Bangladesh.**
(PDF)

**S1 Fig. Pictures collected during the field visit shows diverse types of aquaculture in the study area.**
(PDF)

## Acknowledgments

We would like to thank Sentinel data hub for providing Sentinel data, Department of Fisheries Bangladesh for secondary data. The first author would like to thank local government and local people of Satkhira for conducting this research. I am deeply thankful to Deha, Hitesh, Stanley, Stephan for their support, and encouragement during my research stay at Hokkaido University.

## Author Contributions

**Conceptualization:** Ram Avtar.

**Data curation:** Hafeza Nujaira, Haroon Sajjad.

**Formal analysis:** Hafeza Nujaira, Ali Kharrazi, L. N. Gupta, Haroon Sajjad.

**Funding acquisition:** Pankaj Kumar.

**Investigation:** Ali Kharrazi, L. N. Gupta, Haroon Sajjad.

**Methodology:** Hafeza Nujaira, Ali P. Yunus.

**Project administration:** Ram Avtar.

**Resources:** Hafeza Nujaira, Kumar Arun Prasad, Ali P. Yunus, Ali Kharrazi, Tonni Agustiono Kurniawan.

**Software:** Kumar Arun Prasad.

**Supervision:** Pankaj Kumar, Ram Avtar.

**Validation:** Hafeza Nujaira, Kumar Arun Prasad, Pankaj Kumar, L. N. Gupta, Tonni Agustiono Kurniawan.

**Visualization:** Ali P. Yunus, Tonni Agustiono Kurniawan.

**Writing – original draft:** Hafeza Nujaira.

**Writing – review & editing:** Kumar Arun Prasad, Pankaj Kumar, Ali P. Yunus, Ali Kharrazi, L. N. Gupta, Tonni Agustiono Kurniawan, Haroon Sajjad, Ram Avtar.

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
