## [Decision Letter · Decision Letter 0]

3 Mar 2022

PONE-D-22-00724Quantification of spatio-temporal variation of aquaculture area in Satkhira, Bangladesh: Using Geospatial and social survey dataPLOS ONE

Dear Dr. Avtar,

Thank you for submitting your manuscript to PLOS ONE. After careful consideration, we feel that it has merit but does not fully meet PLOS ONE’s publication criteria as it currently stands. Therefore, we invite you to submit a revised version of the manuscript that addresses the points raised during the review process.

We look forward to receiving your revised manuscript.

Kind regards,

Bijeesh Kozhikkodan Veettil

Academic Editor

PLOS ONE

Journal Requirements:

2. Please include a complete ethics statement in the Methods section, including information on how participants were recruited, whether an IRB was consulted or any permits obtained, and if so, the name of the IRB or authority and the approval number, and whether they approved the study or waived the need for approval. Please also clarify whether the participants provided consent, and if so, how, or whether the IRB waived the need for consent.

3. Please include a complete copy of PLOS’ questionnaire on inclusivity in global research in your revised manuscript. Our policy for research in this area aims to improve transparency in the reporting of research performed outside of researchers’ own country or community. The policy applies to researchers who have travelled to a different country to conduct research, research with Indigenous populations or their lands, and research on cultural artefacts. The questionnaire can also be requested at the journal’s discretion for any other submissions, even if these conditions are not met.  Please find more information on the policy and a link to download a blank copy of the questionnaire here: https://journals.plos.org/plosone/s/best-practices-in-research-reporting. Please upload a completed version of your questionnaire as Supporting Information when you resubmit your manuscript.

4. We note that Figures 1, 3, 6, 7 and 9 in your submission contain map images which may be copyrighted. All PLOS content is published under the Creative Commons Attribution License (CC BY 4.0), which means that the manuscript, images, and Supporting Information files will be freely available online, and any third party is permitted to access, download, copy, distribute, and use these materials in any way, even commercially, with proper attribution. For these reasons, we cannot publish previously copyrighted maps or satellite images created using proprietary data, such as Google software (Google Maps, Street View, and Earth). For more information, see our copyright guidelines: http://journals.plos.org/plosone/s/licenses-and-copyright.

a. You may seek permission from the original copyright holder of Figures Figures 1, 3, 6, 7 and 9 to publish the content specifically under the CC BY 4.0 license.  

Additional Editor Comments:

Major revisions required. One reviewer suggested rejection. Please consider all the reviewer comments when preparing the revised version.

Reviewers' comments:

Reviewer's Responses to Questions

**Comments to the Author**

1. Is the manuscript technically sound, and do the data support the conclusions?

Reviewer #1: Yes

Reviewer #2: No

Reviewer #3: Yes

2. Has the statistical analysis been performed appropriately and rigorously? 

Reviewer #1: Yes

Reviewer #2: No

Reviewer #3: Yes

3. Have the authors made all data underlying the findings in their manuscript fully available?

Reviewer #1: No

Reviewer #2: No

Reviewer #3: Yes

4. Is the manuscript presented in an intelligible fashion and written in standard English?

Reviewer #1: No

Reviewer #2: No

Reviewer #3: Yes

5. Review Comments to the Author

Reviewer #1: The paper is good, but it needs some enhancements:

- Figures quality not good. Improve them.

- English has typos.

- More details about the employed methods can be added.

- Elaborate your motivation, contribution in the introduction section.

-The conclusion must be improved. The authors should focus on their unique work and contributions at first, and they should support their conclusion by numerical results. Then, the limitations of this paper should be discussed. Accordingly, the future work of this paper can be drawn;

Reviewer #2: This study doesn't include a proposal for a paper for the journal Plos One.

The theme is interesting, but it has little data (temporal and spatial) and incomplete statistical evaluation to be published in a scientific journal like a Plos One.

I recommend submitting the work to a symposium, congress or similar.

Reviewer #3: The title of the manuscript is interesting but there are some major concerns need to be addressed are as follow:

1. The novelty of the manuscript is questionable. Authors need to highlight “what is the exact novelty of this manuscript ?” and in a better way, it needs to be presented in enumerate points.

2. There is no description regarding the splitting of the dataset and how they used to evaluate the experiment ? because it is important for another researcher to replicate that experiment.

3. There are some statistical parameters used in this study for the evaluation but there is a description as well as numerical presentation required.

4. What is the experimental testbed used to achieve that output ? It should be hardware and software both required.

5. There are introductory lines required between the section and subsection to make flow for the readers.

6. PLOS authors have the option to publish the peer review history of their article (what does this mean?). If published, this will include your full peer review and any attached files.

Reviewer #1: No

Reviewer #2: No

Reviewer #3: **Yes: **Vikram Puri

---

## [Author Response · Author response to Decision Letter 0]

2 Aug 2022

Dear Editor,

We very much appreciate all the reviewers and editor for encouraging, critical and constructive comments on this manuscript. The comments and suggestions have been extensive and useful to improve the manuscript. We strongly believe that these comments and suggestions have increased the scientific value of the revised manuscript by many folds. We have taken them fully into account in revision. We are submitting the revised version of the manuscript with the suggestion incorporated in the manuscript. The manuscript has been revised as per the comments given by the reviewer, and our responses to all the comments are as follows:

Reviewer #1: The paper is good, but it needs some enhancements:

Reply : Thank you so much for your useful comments and suggestion. We greatly appreciate them for making our manuscript better.

- Figures quality not good. Improve them.

Reply: Thank you very much for your suggestion. We have revised the figures and improved the quality of manuscript.

- English has typos.

Reply: We have revised the manuscript and revised English thoroughly.

- More details about the employed methods can be added.

Reply: Thank you very much for your suggestion. We have revised the methodology and added more details in the revised version of the manuscript. (Line: 158-340)

- Elaborate your motivation, contribution in the introduction section.

Reply: Thank you very much for your suggestion. We have revised the motivation and contribution in the introduction part. (Line: 103-124 )

-The conclusion must be improved. The authors should focus on their unique work and contributions at first, and they should support their conclusion by numerical results. Then, the limitations of this paper should be discussed. Accordingly, the future work of this paper can be drawn;

Reply: Thank you very much for your kind suggestion. We have thoroughly revised the conclusion and included limitations and future scope of this study in the revised version of the discussion (Line: 617-627 )

Reviewer #2: This study doesn't include a proposal for a paper for the journal Plos One.

The theme is interesting, but it has little data (temporal and spatial) and incomplete statistical evaluation to be published in a scientific journal like a Plos One.

I recommend submitting the work to a symposium, congress or similar.

Reply: Thank you very much for your suggestion. We have clearly mentioned motivation and contribution of this manuscript in the introduction part of the manuscript. This study is important for data scarce countries like Bangladesh. This study will provide innovative approach to monitor aquaculture areas using freely available geospatial data to improve aquaculture management. As aquaculture contribute towards GDP it is essential to manage them sustainable. Hope you will find them justified.

Reviewer #3: The title of the manuscript is interesting but there are some major concerns need to be addressed are as follow:

Reply: Thank you so much for your useful comments and suggestion. We greatly appreciate them for making our manuscript better.

1. The novelty of the manuscript is questionable. Authors need to highlight “what is the exact novelty of this manuscript ?” and in a better way, it needs to be presented in enumerate points.

Reply : Thank you so much for your useful suggestion. We have revised the motivation and contribution in the introduction part to reflect the novelty of this manuscript. (Line: 103-124)

2. There is no description regarding the splitting of the dataset and how they used to evaluate the experiment ? because it is important for another researcher to replicate that experiment.

Reply: Thank you very much for your suggestion. We have revised the methodology and added more details in the revised version of the manuscript. It will help other researches to replicate the methodology in other study areas. (Line: 158-340 )

3. There are some statistical parameters used in this study for the evaluation but there is a description as well as numerical presentation required.

Reply: Thank you very much for your suggestion. We have revised the methodology and added numerical formulas that has been used in this study in the revised version of the manuscript. 

4. What is the experimental testbed used to achieve that output ? It should be hardware and software both required.

Reply: Thank you very much for your suggestion. We have included information about softwares used in this study that can other researchers to follow the methodology.

5. There are introductory lines required between the section and subsection to make flow for the readers.

Reply: Thank you very much for your suggestion. We have thoroughly revised the introduction part of the manuscript to make it more coherent.

---

## [Decision Letter · Decision Letter 1]

19 Aug 2022

PONE-D-22-00724R1Quantification of spatio-temporal variation of aquaculture area in Satkhira, Bangladesh: Using Geospatial and social survey dataPLOS ONE

Dear Dr. Avtar,

Thank you for submitting your manuscript to PLOS ONE. After careful consideration, we feel that it has merit but does not fully meet PLOS ONE’s publication criteria as it currently stands. Therefore, we invite you to submit a revised version of the manuscript that addresses the points raised during the review process.

We look forward to receiving your revised manuscript.

Kind regards,

Bijeesh Kozhikkodan Veettil

Academic Editor

PLOS ONE

Journal Requirements:

Reviewers' comments:

Reviewer's Responses to Questions

**Comments to the Author**

1. If the authors have adequately addressed your comments raised in a previous round of review and you feel that this manuscript is now acceptable for publication, you may indicate that here to bypass the “Comments to the Author” section, enter your conflict of interest statement in the “Confidential to Editor” section, and submit your "Accept" recommendation.

Reviewer #2: (No Response)

Reviewer #3: (No Response)

2. Is the manuscript technically sound, and do the data support the conclusions?

Reviewer #2: No

Reviewer #3: Yes

3. Has the statistical analysis been performed appropriately and rigorously? 

Reviewer #2: No

Reviewer #3: Yes

4. Have the authors made all data underlying the findings in their manuscript fully available?

Reviewer #2: No

Reviewer #3: Yes

5. Is the manuscript presented in an intelligible fashion and written in standard English?

Reviewer #2: Yes

Reviewer #3: Yes

6. Review Comments to the Author

Reviewer #2: The text has been significantly improved, but still needs some tweaking.

# I recommend inserting the information obtained, in addition to that mentioned in the article, about secondary data and the questionnaire applied in December 2019.

# I recommend inserting radiometric and atmospheric corrections into Sentinel 2 data.

# I recommend entering how many scenes from sentinel 1 and 2 were used between January 2017 to December 2019

# I also recommend adding the temporal range in 3.1.2, in order to facilitate the reading

# I recommend detailing in 3.3 the pre-processing procedures performed on sentinel data 1 and 2.

# I recommend entering the results of non-parametric statistical analysis in 3.3.2 (referring to Figure 4), and 4.2.1 (referring to Figure 10). Do not forget to include the analyzed methodology of these analyses.

# I recommend not using the kappa index. Since 2011, the Remote Sensing community has avoided using this index. See the article: https://doi.org/10.1080/01431161.2011.552923

# I recommend inserting the MDWI and MNDWI maps to winter and pre-monsoon seasons (Figure 7) and the same season for Figure 8.

Reviewer #3: Authors need to mention in the introduction what techniques are used in this research as well as the novelty of the manuscript mentioned pointwise.

Comment 3 & 4, where authors highlight the statistical parameters formula and software details respectively?

7. PLOS authors have the option to publish the peer review history of their article (what does this mean?). If published, this will include your full peer review and any attached files.

Reviewer #2: No

Reviewer #3: No

---

## [Author Response · Author response to Decision Letter 1]

7 Nov 2022

Dear Editor and reviewers,

We very much appreciate all the reviewers and editor for encouraging, critical and constructive comments on this manuscript. The comments and suggestions have been extensive and useful to improve the manuscript. We strongly believe that these comments and suggestions have increased the scientific value of the revised manuscript by many folds. We have taken them fully into account in revision. We are submitting the revised version of the manuscript with the suggestion incorporated in the manuscript. The manuscript has been revised as per the comments given by the reviewer, and our responses to all the comments are as follows:

Reviewer# 1

No. Comments Responses

1 I recommend inserting the information obtained, in addition to that mentioned in the article, about secondary data and the questionnaire applied in December 2019. We once again thank you very much for your time and constructive feedback. Following your suggestion, we now incorporate the questionnaire and survey response transcript as supplementary material with this paper. We also mentioned it in the section 4.3. The questionnaire survey form and results based on questionnaire survey is provided in supplementary file S1 and supplementary file S2, respectively. 

 2 I recommend inserting radiometric and atmospheric corrections into Sentinel 2 data.

 We added the following: 

See line 174-187

“We obtained the pre-processed surface reflectance Sentinel-2 L2A data from GEE through scihub. They were initially computed by running Sen2Cor processor, consists in scene classification and atmospheric correction applied to Level-1C orthoimage product. Atmospheric correction in Sen2Cor is performed using a set of look-up tables generated via libRadtran. The aerosol type and visibility or optical thickness of the atmosphere is derived using the Dense Dark Vegetation (DDV) algorithm. Clouds if any present in the scene are removed by using COPERNICUS/S2_CLOUD_PROBABILITY algorithm. A total of 380 Sentinel-2 scenes available between 2017 January and 2019 December were processed in GEE for this study (See Supplementary file S3)”.

3 I recommend entering how many scenes from sentinel 1 and 2 were used between January 2017 to December 2019 Thank you. We added this information in revised manuscript.

See Line 165-170 A total of 56 Sentinel-1 scenes available between 2017 January and 2019 December were used in this study (See Supplementary file S3).

4 I also recommend adding the temporal range in 3.1.2, in order to facilitate the reading 

 Added:

See Line 178-180: A total of 380 Sentinel-2 scenes available between 2017 January and 2019 December were processed in GEE for NDWI and MNDWI extraction (See supplementary file F3).

5 I recommend detailing in 3.3 the pre-processing procedures performed on sentinel data 1 and 2. We revised the section 3.3 for clarity. Processing codes for NDWI and MNDWI are now provided in the data availability section. 

See the code for MNDWI extraction from S2 L2A in GEE: Supplementary file S4

Pre-processing procedure for S1 and S2 are now presented in section 3.1

6 I recommend entering the results of non-parametric statistical analysis in 3.3.2 (referring to Figure 4), and 4.2.1 (referring to Figure 10). Do not forget to include the analyzed methodology of these analyses. Thank you very much. We added the details of non-parametric tests – referring to fig 4 in section 3.3.2. See Line 262-275” 

For each shape metric we used, the non-parametric Mann Whitney U test (also known as Wilcoxon rank sum test) was conducted to see any significant difference in the metric values between the 5 sites. The analysis was done in R language v. 4.1.3 with the ‘wilcox_test()’ function in the package ‘rstatix’. The test was conducted for each combination of sites, with null hypothesis that there is no shift in the distribution of site 1 and 2, and alternative hypothesis is that group 1 is shifted to the left of group 2. We noticed that for the area metrics, Satkhira Sadar had significantly larger ponds than Assassuni (p < 0.001), Kaligange (p = 0.009) and Debhata (p = 0.026). However, Satkhira Sadar’s pond perimeter was only significantly larger than those of Assassuni (p = 0.039). For the compactness metrics, Debhata had significantly higher values than Assassuni (p < 0.001) and Kaligange (p <0.001), and similarly Satkhira Sadar had significantly higher values than Assassuni (p = 0.006) and Kaligange (p = 0.008). For the P2A metric, Kaligange was higher than Debhata (p < 0.001) and Satkhira Sadar (p = 0.08), and Assassuni was also higher than Debhata (p < 0.001) and Satkhira Sadar (p = 0.06). All reported p-values are adjusted for multiple comparison using the Holm method”.

Because of limited datapoints, no non-parametric test was conducted for Fig.10.

7 # I recommend not using the kappa index. Since 2011, the Remote Sensing community has avoided using this index. See the article: https://doi.org/10.108 0/01431161.2011.552923. 

We agree with the reviewer here. As reported in Stehmen and Foody, 2019 (Remote Sensing of Environment).. “Liu et al. (2007) showed that kappa was highly correlated with overall accuracy, which is evident from Eq. (24), so reporting both measures is redundant”. ………. ”………….. it may cause no serious harm if you have it and pay little attention to it, but it does not fulfil a necessary function.”. https://doi.org/10.1016/j.rse.2019.05.018

Thus, the accuracy of our results can be noted from the overall accuracy and kappa values were eliminated. 

8 I recommend inserting the NDWI and MNDWI maps to winter and pre-monsoon seasons (Figure 7) and the same season for Figure 8. Thank you very much for your kind suggestion. As we have generated NDWI and MNDWI data using Sentinel-2 data in google earth engine. As Bangladesh is a tropical country and Sentinel-2 was full of clouds during the pre-monsoon and monsoon months therefore, we didn’t use in this analysis. However, we checked the trend of the aquaculture area using Sentinel-1 (SAR) data to overcome the limitations of clouds. 

 Reviewer# 2

No Comments Responses

1 Authors need to mention in the introduction what techniques are used in this research as well as the novelty of the manuscript mentioned pointwise. Thanks for the time and constructive suggestions. Following your comment, we revised the introduction section for clarity.

See Line 101-120

Although Bangladesh plays an important role in the aquaculture sector and contributes significantly to the national GDP, the lack of up-to-date, explicit and continuous spatial knowledge about aquaculture imposes a great hurdle in its sustainable management. Here, we employed multi-temporal Sentinel-1 and Sentinel-2 images from 2017-2019 to track the changes in aquaculture productivity in the Satkhira district, Bangladesh. This study focuses on providing a holistic picture of factors responsible for the trend in aquaculture, its related opportunities and challenges for people in the Satkhira district. The main objectives of this study are: (a) Monitor the spatio-temporal extent of aquaculture ponds from 2017 to 2019 in Satkhira district using an integrated geospatial and field approach; and (b) provide detailed information on socio-economic perspectives on aquaculture in Satkhira, Bangladesh, to enable a more sustainable and profitable management. While the first objective can be achieved by quantitative remote sensing approach, the second objective is qualitative and used key informant interviews with the relevant stakeholders in the region. T

This study will be useful to identify not only the spatio-temporal variation but as well problems associated with the aquaculture industry. It also proposes a possible management solution that can be beneficial for common farmers and other stakeholders, such as government and NGOs. In addition, this research will play an important role for the government in achieving the SDG goals. In particular, mapping and quantification of existing aquaculture areas can contribute to food security (SDG 2); clean water and sanitation (SDG 6.0), economic growth and better livelihoods (SDG 8); and sustainable consumption (SDG 12), to name a few.

2 Comment 3 & 4, where authors highlight the statistical parameters formula and software details respectively? Thank you very much. We added the details of non-parametric tests – referring to fig 4 in section 3.3.2. See Line 259-272” 

For each shape metric we used, the non-parametric Mann Whitney U test (also known as Wilcoxon rank sum test) was conducted to see any significant difference in the metric values between the 5 sites. The analysis was done in R language v. 4.1.3 with the ‘wilcox_test()’ function in the package ‘rstatix’. The test was conducted for each combination of sites, with null hypothesis that there is no shift in the distribution of site 1 and 2, and alternative hypothesis is that group 1 is shifted to the left of group 2. We noticed that for the area metrics, Satkhira Sadar had significantly larger ponds than Assassuni (p < 0.001), Kaligange (p = 0.009) and Debhata (p = 0.026). However, Satkhira Sadar’s pond perimeter was only significantly larger than those of Assassuni (p = 0.039). For the compactness metrics, Debhata had significantly higher values than Assassuni (p < 0.001) and Kaligange (p <0.001), and similarly Satkhira Sadar had significantly higher values than Assassuni (p = 0.006) and Kaligange (p = 0.008). For the P2A metric, Kaligange was higher than Debhata (p < 0.001) and Satkhira Sadar (p = 0.08), and Assassuni was also higher than Debhata (p < 0.001) and Satkhira Sadar (p = 0.06). All reported p-values are adjusted for multiple comparison using the Holm method”.

We additionally explained in section 3.1 and 3.3 the pre-processing methods carried with the satellite images and software employed.

Dear Editor and reviewers,

We very much appreciate all the reviewers and editor for encouraging, critical and constructive comments on this manuscript. The comments and suggestions have been extensive and useful to improve the manuscript. We strongly believe that these comments and suggestions have increased the scientific value of the revised manuscript by many folds. We have taken them fully into account in revision. We are submitting the revised version of the manuscript with the suggestion incorporated in the manuscript. The manuscript has been revised as per the comments given by the reviewer, and our responses to all the comments are as follows:

Reviewer# 1

No. Comments Responses

1 I recommend inserting the information obtained, in addition to that mentioned in the article, about secondary data and the questionnaire applied in December 2019. We once again thank you very much for your time and constructive feedback. Following your suggestion, we now incorporate the questionnaire and survey response transcript as supplementary material with this paper. We also mentioned it in the section 4.3. The questionnaire survey form and results based on questionnaire survey is provided in supplementary file S1 and supplementary file S2, respectively. 

 2 I recommend inserting radiometric and atmospheric corrections into Sentinel 2 data.

 We added the following: 

See line 174-187

“We obtained the pre-processed surface reflectance Sentinel-2 L2A data from GEE through scihub. They were initially computed by running Sen2Cor processor, consists in scene classification and atmospheric correction applied to Level-1C orthoimage product. Atmospheric correction in Sen2Cor is performed using a set of look-up tables generated via libRadtran. The aerosol type and visibility or optical thickness of the atmosphere is derived using the Dense Dark Vegetation (DDV) algorithm. Clouds if any present in the scene are removed by using COPERNICUS/S2_CLOUD_PROBABILITY algorithm. A total of 380 Sentinel-2 scenes available between 2017 January and 2019 December were processed in GEE for this study (See Supplementary file S3)”.

3 I recommend entering how many scenes from sentinel 1 and 2 were used between January 2017 to December 2019 Thank you. We added this information in revised manuscript.

See Line 165-170 A total of 56 Sentinel-1 scenes available between 2017 January and 2019 December were used in this study (See Supplementary file S3).

4 I also recommend adding the temporal range in 3.1.2, in order to facilitate the reading 

 Added:

See Line 178-180: A total of 380 Sentinel-2 scenes available between 2017 January and 2019 December were processed in GEE for NDWI and MNDWI extraction (See supplementary file F3).

5 I recommend detailing in 3.3 the pre-processing procedures performed on sentinel data 1 and 2. We revised the section 3.3 for clarity. Processing codes for NDWI and MNDWI are now provided in the data availability section. 

See the code for MNDWI extraction from S2 L2A in GEE: Supplementary file S4

Pre-processing procedure for S1 and S2 are now presented in section 3.1

6 I recommend entering the results of non-parametric statistical analysis in 3.3.2 (referring to Figure 4), and 4.2.1 (referring to Figure 10). Do not forget to include the analyzed methodology of these analyses. Thank you very much. We added the details of non-parametric tests – referring to fig 4 in section 3.3.2. See Line 262-275” 

For each shape metric we used, the non-parametric Mann Whitney U test (also known as Wilcoxon rank sum test) was conducted to see any significant difference in the metric values between the 5 sites. The analysis was done in R language v. 4.1.3 with the ‘wilcox_test()’ function in the package ‘rstatix’. The test was conducted for each combination of sites, with null hypothesis that there is no shift in the distribution of site 1 and 2, and alternative hypothesis is that group 1 is shifted to the left of group 2. We noticed that for the area metrics, Satkhira Sadar had significantly larger ponds than Assassuni (p < 0.001), Kaligange (p = 0.009) and Debhata (p = 0.026). However, Satkhira Sadar’s pond perimeter was only significantly larger than those of Assassuni (p = 0.039). For the compactness metrics, Debhata had significantly higher values than Assassuni (p < 0.001) and Kaligange (p <0.001), and similarly Satkhira Sadar had significantly higher values than Assassuni (p = 0.006) and Kaligange (p = 0.008). For the P2A metric, Kaligange was higher than Debhata (p < 0.001) and Satkhira Sadar (p = 0.08), and Assassuni was also higher than Debhata (p < 0.001) and Satkhira Sadar (p = 0.06). All reported p-values are adjusted for multiple comparison using the Holm method”.

Because of limited datapoints, no non-parametric test was conducted for Fig.10.

7 # I recommend not using the kappa index. Since 2011, the Remote Sensing community has avoided using this index. See the article: https://doi.org/10.108 0/01431161.2011.552923. 

We agree with the reviewer here. As reported in Stehmen and Foody, 2019 (Remote Sensing of Environment).. “Liu et al. (2007) showed that kappa was highly correlated with overall accuracy, which is evident from Eq. (24), so reporting both measures is redundant”. ………. ”………….. it may cause no serious harm if you have it and pay little attention to it, but it does not fulfil a necessary function.”. https://doi.org/10.1016/j.rse.2019.05.018

Thus, the accuracy of our results can be noted from the overall accuracy and kappa values were eliminated. 

8 I recommend inserting the NDWI and MNDWI maps to winter and pre-monsoon seasons (Figure 7) and the same season for Figure 8. Thank you very much for your kind suggestion. As we have generated NDWI and MNDWI data using Sentinel-2 data in google earth engine. As Bangladesh is a tropical country and Sentinel-2 was full of clouds during the pre-monsoon and monsoon months therefore, we didn’t use in this analysis. However, we checked the trend of the aquaculture area using Sentinel-1 (SAR) data to overcome the limitations of clouds. 

 Reviewer# 2

No Comments Responses

1 Authors need to mention in the introduction what techniques are used in this research as well as the novelty of the manuscript mentioned pointwise. Thanks for the time and constructive suggestions. Following your comment, we revised the introduction section for clarity.

See Line 101-120

Although Bangladesh plays an important role in the aquaculture sector and contributes significantly to the national GDP, the lack of up-to-date, explicit and continuous spatial knowledge about aquaculture imposes a great hurdle in its sustainable management. Here, we employed multi-temporal Sentinel-1 and Sentinel-2 images from 2017-2019 to track the changes in aquaculture productivity in the Satkhira district, Bangladesh. This study focuses on providing a holistic picture of factors responsible for the trend in aquaculture, its related opportunities and challenges for people in the Satkhira district. The main objectives of this study are: (a) Monitor the spatio-temporal extent of aquaculture ponds from 2017 to 2019 in Satkhira district using an integrated geospatial and field approach; and (b) provide detailed information on socio-economic perspectives on aquaculture in Satkhira, Bangladesh, to enable a more sustainable and profitable management. While the first objective can be achieved by quantitative remote sensing approach, the second objective is qualitative and used key informant interviews with the relevant stakeholders in the region. T

This study will be useful to identify not only the spatio-temporal variation but as well problems associated with the aquaculture industry. It also proposes a possible management solution that can be beneficial for common farmers and other stakeholders, such as government and NGOs. In addition, this research will play an important role for the government in achieving the SDG goals. In particular, mapping and quantification of existing aquaculture areas can contribute to food security (SDG 2); clean water and sanitation (SDG 6.0), economic growth and better livelihoods (SDG 8); and sustainable consumption (SDG 12), to name a few.

2 Comment 3 & 4, where authors highlight the statistical parameters formula and software details respectively? Thank you very much. We added the details of non-parametric tests – referring to fig 4 in section 3.3.2. See Line 259-272” 

For each shape metric we used, the non-parametric Mann Whitney U test (also known as Wilcoxon rank sum test) was conducted to see any significant difference in the metric values between the 5 sites. The analysis was done in R language v. 4.1.3 with the ‘wilcox_test()’ function in the package ‘rstatix’. The test was conducted for each combination of sites, with null hypothesis that there is no shift in the distribution of site 1 and 2, and alternative hypothesis is that group 1 is shifted to the left of group 2. We noticed that for the area metrics, Satkhira Sadar had significantly larger ponds than Assassuni (p < 0.001), Kaligange (p = 0.009) and Debhata (p = 0.026). However, Satkhira Sadar’s pond perimeter was only significantly larger than those of Assassuni (p = 0.039). For the compactness metrics, Debhata had significantly higher values than Assassuni (p < 0.001) and Kaligange (p <0.001), and similarly Satkhira Sadar had significantly higher values than Assassuni (p = 0.006) and Kaligange (p = 0.008). For the P2A metric, Kaligange was higher than Debhata (p < 0.001) and Satkhira Sadar (p = 0.08), and Assassuni was also higher than Debhata (p < 0.001) and Satkhira Sadar (p = 0.06). All reported p-values are adjusted for multiple comparison using the Holm method”.

We additionally explained in section 3.1 and 3.3 the pre-processing methods carried with the satellite images and software employed.

---

## [Editor Report · Decision Letter 2]

9 Nov 2022

Quantifying spatio-temporal variation in aquaculture production areas in Satkhira, Bangladesh using geospatial and social survey

PONE-D-22-00724R2

Dear Dr. Avtar,

We’re pleased to inform you that your manuscript has been judged scientifically suitable for publication and will be formally accepted for publication once it meets all outstanding technical requirements.

Kind regards,

Bijeesh Kozhikkodan Veettil

Academic Editor

PLOS ONE
---

## [Editor Report · Acceptance letter]

17 Nov 2022

PONE-D-22-00724R2 

Quantifying spatio-temporal variation in aquaculture production areas in Satkhira, Bangladesh using geospatial and social survey 

Dear Dr. Avtar:

I'm pleased to inform you that your manuscript has been deemed suitable for publication in PLOS ONE. Congratulations! Your manuscript is now with our production department. 

Kind regards, 

on behalf of

Dr. Bijeesh Kozhikkodan Veettil 

Academic Editor

PLOS ONE